# A UNIFIED LIGHTWEIGHT COMPLEX SCENES-ORIENTED NETWORK FOR INFRARED AND VISIBLE IMAGE FUSION

## ABSTRACT

Existing infrared and visible image fusion (IVIF) techniques typically integrate the useful information from different modalities within the ideal conditions. Nevertheless, current state-of-the-art IVIF methods are ineffective when facing complex scene interferences such as bad weather, low light, and high noise, and they typically need to be used in conjunction with other de-interference baselines, which inevitably resulting in the high memory costs and error accumulation, thus yielding sub-optimal fusion results. To address these challenges, We propose a unified lightweight real-time IVIF network for multiple complex scenes. We conducted a theoretically thorough analysis of modal degradations in the frequency domain, leveraging the complementary strengths of both modalities to enhance network learning. Our method facilitates the extraction of critical features even amidst significant pixel interference. For reconstructing fusion results, we introduce a spatial domain branching strategy which significantly improves the local detail resolution, thereby mitigating potential omissions from frequency domain analysis. Extensive qualitative and quantitative experiments demonstrate that our framework excels in handling multiple complex scenes, while maintaining real-time computational efficiency for prompt image processing applications.

## 1 INTRODUCTION

The objective of infrared and visible image fusion (IVIF) is to amalgamate valuable information from diverse modalities to attain a more comprehensive and precise representation of the scene (Zhao et al., 2024; Li & Wu, 2024; Zhang et al., 2021; Ma et al., 2019a; Zhang & Demiris, 2023; Zhao et al., 2023b; Liu et al., 2023b). The technique is widely used in real-world application scenes such as object detection (Wang et al., 2023; Bochkovskiy et al., 2020; Ma et al., 2023), semantic segmentation (Li et al., 2023c; Chen et al., 2017) and autonomous driving (Xiao et al., 2020).

In recent years, the main research in IVIF has focused on ideal fusion scenes, which can be mainly categorised into traditional algorithms (Li et al., 2024a; Zhou et al., 2023; Zhao et al., 2020; Chen et al., 2022; Nie et al., 2021) and deep learning based approaches (Liu et al., 2024b; 2022; Li et al., 2023d; Xu et al., 2020; Huang et al., 2022; Tang et al., 2022a). Traditional methods typically represent the source image at multiple scales and extract multi-modality features across different scale levels. For example, PFF (Zhou et al., 2023) proposed a multi-scale fusion framework for IVIF based on bio-visual inspirations. MCSCM (Luo et al., 2023) proposed an IVIF framework based on Multi-State contextual hidden Markov Model. However, these methods exhibit limited generalization capabilities and demand substantial computational resources. DL-based methods, particularly Convolutional Neural Networks (CNN) and Transformers, have demonstrated superior performance in IVIF tasks compared to traditional methods. For example, U2Fusion (Xu et al., 2020) designed a unified framework for diverse image fusion tasks. Swinfusion (Ma et al., 2022) introduced Swin Transformer to the fusion task, designing a fusion framework capable of capturing long range contextual relationships. In addition, some algorithms (Zhao et al., 2023a) combine Transformer and CNN modules to facilitate specific feature extraction for effective learning of global and local features. Despite the demonstrated effectiveness of the self-attention mechanism in Transformers for global feature extraction, its complexity scales quadratically with the size of the input features. This constraint hampers its widespread deployment in foundational vision tasks.

Figure 1: Example of fused results in two complex scenes. In the rain scene, we show the deraining images and the corresponding high frequency and fusion results respectively. In low-light scene, the fusion result is obtained by fusing visible images processed using a low-light enhancement algorithm. We use restormer (Zamir et al., 2022) for deraining, retinexformer (Cai et al., 2023) for enhancement, and cddfuse (Zhao et al., 2023a) for fusion.

The aforementioned algorithms (Li et al., 2024a; Zhou et al., 2023; Li et al., 2023d; Xu et al., 2020) typically achieve the primary goals of IVIF. However, they often fail to extend effectively to complex scenes. A prevalent strategy for complex scenes involves integrating an image restoration model, which removes interfering features through image pre-processing prior to fusion. Moreover, several fusion architectures designed for complex scenes have been developed (Xu et al., 2023; Xie et al., 2023; Li et al., 2024b). These architectures generally utilize a two-stage learning process, initially addressing image restoration and fusion tasks separately, and then optimizing them interactively. Unfortunately, these methods face three significant challenges: 1) **Error accumulation can occur**, where residual interfering pixels or detail loss from the image restoration stage may propagate to the fusion stage. Existing fusion models, often trained under ideal conditions, may misinterpret these disruptive features as valuable, potentially exacerbating them. For instance, in deraining tasks, raindrop textures that obscure scene information might be mistakenly enhanced as salient features during fusion, as shown in Figure 1(a). 2) **Adding irrelevant or erroneous features**. IVIF seeks to harness the complementary strengths of different modalities. In low-light conditions, the visible modality captures minimal information and relies heavily on the infrared image. Preprocessing with low-light enhancement algorithms may inadvertently introduce spurious or erroneous features from the visible image, as illustrated in Figure 1(b). 3) **Inference costs increase** due to added model parameters and computational complexity. Practical applications in real-world scenes demand that algorithms be highly efficient. Given the scarcity of image fusion architectures suitable for complex scenes, we consider, "**Whether it is feasible to develop a unified framework that supports high-quality and real-time fusion, rather than depending on a two-stage processing approach**."

The answer is yes. In the context of an end-to-end unified framework designed for complex scenes, our primary focus is to enable the fusion network to effectively differentiate between interfering and valuable features. Previous fusion algorithms typically rely on extensive multi-modality data from ideal scenes to train networks to extract salient features from different modalities. When faced with interference, image reconstruction is often used as a precursor to obtain relatively clean source images for subsequent fusion. Our approach moves beyond the traditional focus on merely learning clear features typical of ideal scenes. While learning the original scene information from a limited set of pertinent features may result in some data loss compared to pristine source images, it is essential to recognize the inherent redundancy within the images themselves (He et al., 2022). Here, we can conceive the interfering pixels as masks that obscure the clear features. For instance, rain lines can be viewed as masks, as they obscure scene details during rainy conditions. Previous studies have shown that noise also functions as a mask (Delord, 1998). Similarly, in dark or overexposed scenes, low-light regions behave like masks, with pixel values ranging from 0 to 1, concealing details that would be visible under normal lighting. Thus, due to the redundancy inherent in images, we can effectively reconstruct scene information even when certain details are missing. With this understanding, we believe that the design of an end-to-end framework should prioritize extracting key information from interference first and then reconstructing the scene based on that information, rather than recovering the scene first and then extracting features. Furthermore, discarding pixel redundancy and ideal fusion environment, the network can prioritize learning the most significant and complementary pixel information from various modalities. The above concepts provides theoretical underpinning for our approach.

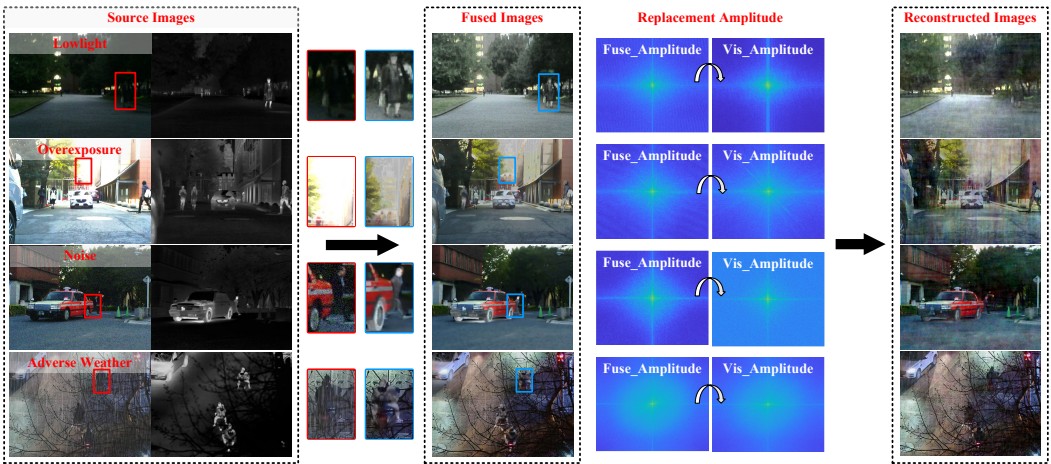

Figure 2: Fusion results of the proposed algorithm in different complex scenes. Fuse-Amplitude and Vis-Amplitude represent the amplitudes of the fused and visible images, respectively. The final column shows the image reconstructed by combining the amplitude of the fused image with the phase of the visible image through the Fourier inverse transform.

We present a fast, robust, and unified IVIF framework that integrates image restoration and fusion. Figure 2 illustrates the effectiveness of the proposed model across four complex scenes, demonstrating its ability to adeptly extract feature information amidst interference. It's worth noting that our algorithm completes the fusion process for a $640 \times 480$ image in just $0.033$ seconds. In addition, by replacing the amplitude of the source image with that of the fusion result, we found that the reconstructed image significantly reduces most interfering pixels. Some studies (Li et al., 2023a; Yu et al., 2022) have demonstrated that image degradation primarily affects the amplitude spectrum. Consequently, proposed framework effectively recovers the amplitude of degraded scenes, leading to high-quality fusion. Our contributions are summarised below:

- We proposed a unified framework for real-time IVIF in complex scenes, enabling high-quality fusion with limited computational resources. To the best of our knowledge, this is the first work of addressing IVIF in complex scenes from a frequency domain perspective.

- We proposed a multi-modality interactive guidance mechanism within the Fourier domain. This strategy efficiently extracts and restores useful features from degraded pixels by leveraging the complementary strengths of different modalities.

- Our framework achieves superior image fusion quality in diverse complex conditions such as rain, overexposure, low-light, and noise. Extensive experiments confirm that our method outperforms state-of-the-art methods while requiring fewer computational resources.

## 2 RELATED WORK

**Infrared and Visible Image Fusion.** Current IVIF algorithms can be categorized into three types: autoencoder (AE)-based models (Ma et al., 2022; Tang et al., 2023a; Liu et al., 2021; Xu et al., 2022), generative adversarial network (GAN)-based models (Liu et al., 2022; Le et al., 2022; Ma et al., 2020), and algorithmic unfolding models (Li et al., 2023b; Deng & Dragotti, 2020). The core idea of AE-based IVIF algorithms (Tang et al., 2023b) revolves around achieving multi-modality information extraction and fusion through learning a compact representation of the image and its subsequent restoration. The GAN-based IVIF algorithm (Ma et al., 2019b) implements multiple multi-modality information extraction mainly through adversarial training of generator and discriminator. Algorithmic unfolding models (Li et al., 2023b) iteratively adjust parameters to better fit the data and optimize the objective function. However, most of the mentioned algorithms (Ma et al., 2022; Tang et al., 2023a; Liu et al., 2022; Li et al., 2023b; Tang et al., 2023b) are designed under ideal fusion scenes, lacking inherent knowledge of complex scenes with multiple disturbances.

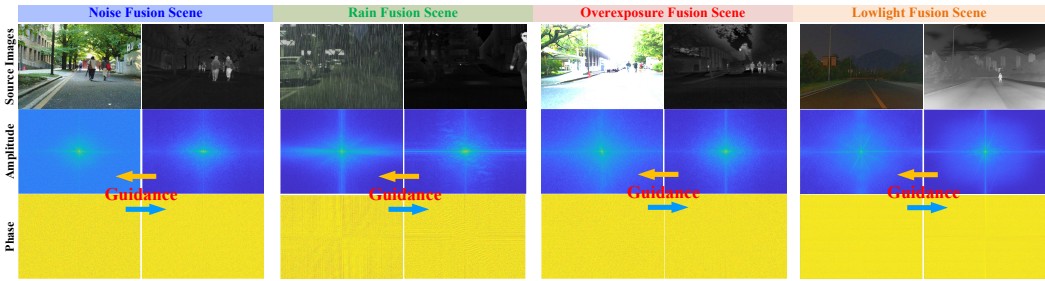

Figure 3: Motivation. Multi-modality interaction guidance mechanism. We use the infrared amplitude to guide the visible amplitude, and the visible phase to guide the infrared phase.

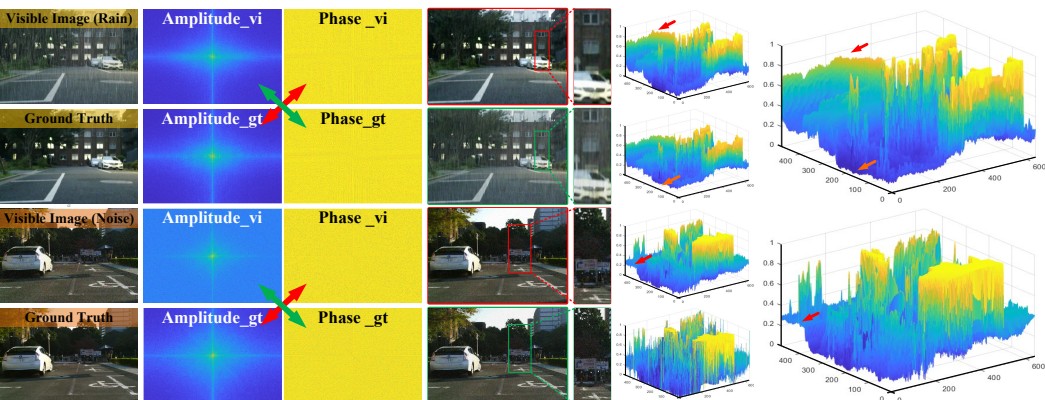

Figure 4: Motivation. The second and third columns show the visualized results of the amplitude and phase spectra of different source images. The fourth column shows the reconstructed image after combining the amplitude and phases of different source images. The sixth column presents a mesh surface map derived from the source image to its left, while the seventh column displays a mesh surface map based on the ground truth.

Therefore, our aim is to explore a unified framework for image recovery and fusion while minimizing computational complexity without compromising performance.

**Frequency Domain Learning.** Learning in frequency domain enhances network interpretability (Lin et al., 2023; Cai et al., 2021; Yang & Soatto, 2020; Suvorov et al., 2022), and improving performance as demonstrated in various visual tasks (Mao et al., 2021; Li et al., 2023a; Yu et al., 2022; Pham et al., 2021; Liu et al., 2023a). Amplitude represents the intensity or energy of individual frequency components within an image. It quantifies how much each frequency component, which ranges from fine details to broader structural features, contributes to the overall image. Phase, on the other hand, encodes the positional information of these frequency components, describing their relative spatial arrangement within the image. Some studies have explored the correlation between frequency characteristics and degraded images, revealing that factors like haze and low-light primarily affect the amplitude of the image (Li et al., 2023a; Yu et al., 2022). Utilizing the Fourier domain aids the network in pinpointing interfering pixels and enhancing the recovery of clear image details. The frameworks mentioned (Li et al., 2023a; Yu et al., 2022; Song et al., 2022) focused on learning in the frequency domain for single-modal tasks and did not extend to multi-modality image processing. Our framework takes a step further by tailoring the frequency prior for both joint IVIF and image restoration tasks for the first time.

## 3 PROPOSED METHOD

**Motivation.** Our proposed framework is inspired by the unique interactions between the amplitude and phase components in the Fourier domain of visible and infrared images under complex scene

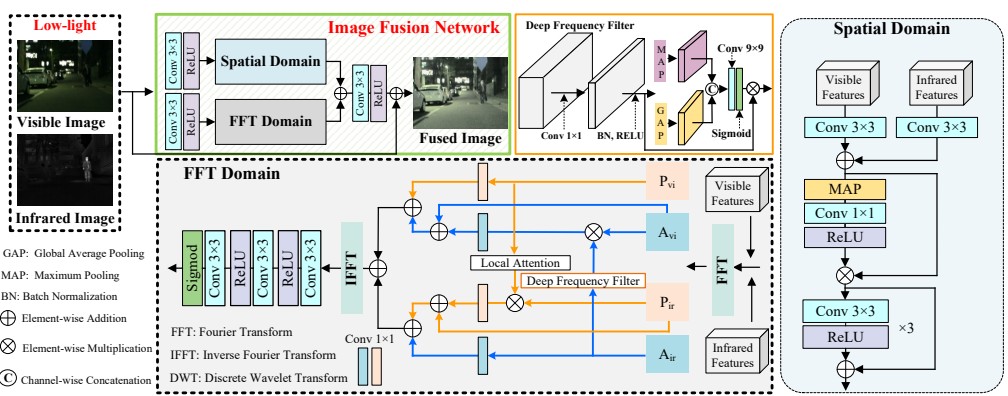

Figure 5: Overview of the proposed unified network for infrared and visible image fusion.

conditions. The infrared spectrum's superior ability to penetrate obscurants like fog, snow, and haze results in images with greater clarity and stability compared to visible light images in non-ideal conditions, such as adverse weather or low-light scenes. To leverage the advantages of infrared imaging, our method uses the amplitude of infrared images to guide the recovery of the visible image's amplitude. However, infrared images often lack intricate texture details; therefore, we employ the phase information from visible images to assist in extracting phase data from infrared images. Importantly, the phase captures only the positional information of pixels, ensuring that degradation information is not transferred to the infrared modality. As demonstrated in Figure 3, this mutual guidance between infrared amplitude and visible phase enables the extraction of richer and more valuable features from both modalities.

In weak interference situations, the amplitude and phase of the visible image may provide overall better information than the IR image. Conversely, in very dark or severely overexposed regions, the infrared image may contain more detail than the visible image, offering superior scene information. In both scenes, our proposed guidance mechanism remains effective. It is important to note that the multi-modality mutual guidance mechanism does not completely replace the information from any specific modality in the frequency domain; rather, it focuses on effectively utilizing the complementary information from both modalities. Additionally, our bootstrapping approach operates on the frequency components in the frequency domain, rather than directly manipulating pixel values in the spatial domain. The goal of this bootstrapping is to better preserve and enhance the periodic modes in the frequency spectrum, rather than to propagate pixel values directly.

Our observations from Figure 4 reveal that amplitude differences are primarily responsible for distinguishing between degraded and clear images, while the phase information remains relatively stable. Thus, reconstructing images with a combination of clear amplitude and original phase can preserve the scene's core information. Still, solely relying on amplitude reconstruction can lead to detail loss, as evident in our mesh surface map (Figure 4). Therefore, our network also integrates spatial domain information to compensate for and enhance textural details, adopting a dual-domain approach for more effective and comprehensive image fusion in complex environments.

## 3.1 IMAGE FUSION NETWORK

Our framework is both simple and effective. Initially, the source images are input into the FFT domain module and the spatial domain module, respectively, to acquire feature maps of different domains. These maps are then summed up, and the final fusion result is obtained by combining the source image information with the feature maps. The model's framework is illustrated in Figure 5.

**Frequency Domain Module.** Learning pixel information in the frequency domain provides an inherent prior for the network (Yang & Soatto, 2020; Suvorov et al., 2022; Li et al., 2023a; Yu et al., 2022). In images affected by adverse weather, low-light, and other disturbances, the representation space of degraded pixels is typically confined to the image's amplitude. In addition, compared to the

amplitude spectrum, the phase indicates the relative position or offset of the signal, predominantly containing structural information of the image. We further extend this Fourier framework knowledge (Li et al., 2023a; Yu et al., 2022) to multi-modality image processing.

The Fourier transform has been widely employed across various fields as an efficient tool for analyzing the frequency components of an image. Given an input image $I_{\text{in}} \in \mathbb{R}^{H \times W \times C}$, where $I_{\text{in}}$ can be denoted $I_{\text{ir}}(I_{\text{vi}})$ to represent infrared(visible) source images, the Fourier transform transforms it into the complex component $F(I_{\text{in}})$ of the frequency domain space,

$$F(I_{\text{in}})(u, v) = \sum_{h=0}^{H-1} \sum_{w=0}^{W-1} I_{\text{in}}(h, w) e^{-j2\pi \left( \frac{h}{H} u + \frac{w}{W} v \right)} \tag{1}$$

The frequency domain feature $F(I_{\text{in}})$ can be further expressed as:

$$F(I_{\text{in}}) = R(I_{\text{in}}) + jI(I_{\text{in}}) \tag{2}$$

where $R(I_{\text{in}})$ and $I(I_{\text{in}})$ denote the real and imaginary parts of $F(I_{\text{in}})$, respectively. The phase spectrum $P(I_{\text{in}})$ and the amplitude spectrum $A(I_{\text{in}})$ can be denoted, respectively, as

$$P(I_{\text{in}})(u, v) = \arctan \left[ \frac{I(I_{\text{in}})(u, v)}{R(I_{\text{in}})(u, v)} \right] \tag{3}$$

$$A(I_{\text{in}})(u, v) = \left[ R^2(I_{\text{in}})(u, v) + I^2(I_{\text{in}})(u, v) \right]^{1/2} \tag{4}$$

In order to enhance the metastable frequency components and suppress the unfavourable frequency components of the latent space for generalisation, we use deep frequency filtering (Lin et al., 2023) to generate the attention map $\text{Atten}_1(F_{\text{ir}})$ that guides the recovery of the visible amplitudes,

$$\text{Atten}_1(F_{\text{ir}}) = \sigma(\text{Conv}_{7 \times 7}([\text{MAP}(A(F_{\text{ir}})), \text{GAP}(A(F_{\text{ir}}))])) \tag{5}$$

where $F_{ir}$ and $F_{vi}$ denote the feature maps obtained after $3 \times 3$ convolution and RELU of the infrared and visible images respectively, $\sigma(\cdot)$ denotes the sigmoid function, $[\cdot, \cdot]$ denotes the concatenation operation, $\text{MAP}(\cdot)$ and $\text{GAP}(\cdot)$ denote the maximum pooling and global average pooling operations, respectively. $\text{Conv}_{7 \times 7}(\cdot)$ denotes the convolution layer with the kernel size of 7. To better extract the weak texture information into the infrared image, we use local attention to generate the attention map $\text{Atten}_2(F_{\text{vi}})$ that guides the infrared phase recovery,

$$\text{Atten}_2(F_{vi}) = \sigma(\text{MAP}(P(F_{vi}))) \tag{6}$$

Subsequently, the output amplitude feature $\tilde{A}(F_{vi})$ and phase feature $\tilde{P}(F_{ir})$ can be obtained by mutual guidance of the attention maps,

$$\tilde{A}(F_{vi})(u, v) = \text{Conv}_{1 \times 1}([\text{Atten}_1(F_{ir})(u, v) \otimes A(F_{vi})(u, v)]) + A(F_{vi})(u, v) \tag{7}$$

$$\tilde{P}(F_{ir})(u, v) = \text{Conv}_{1 \times 1}([\text{Atten}_2(F_{vi})(u, v) \otimes P(F_{ir})(u, v)]) + P(F_{ir})(u, v) \tag{8}$$

where $\otimes$ represents element-wise multiplication. After re-transforming the amplitude and phase features into real and imaginary parts, we summed these components across the different modalities. Finally, they are transformed into spatial domain features using the Fourier inverse transformation. For the specific flow of the frequency domain module, please refer to Figure 5.

**Spatial Domain Module.** The spatial domain information compensates for details overlooked in frequency domain learning, requiring only the capture of sparse and significant pixel information. MAP is a simple and effective tool for this purpose. MAP selects features with the highest response in each window while discarding weaker details. This mechanism enables our model to avoid producing redundant features and conserves computational resources.

Firstly, we increase the channel number of the infrared and visible image features to 64 through $3 \times 3$ convolution operation, to obtain $\tilde{F}_{ir}$ and $\tilde{F}_{vi}$. These features are then summed and input into the MAP model. These salient features undergo further refinement and restructuring via a $1 \times 1$ convolution, yielding the sparse feature attention map $\text{Atten}_3(\tilde{F}_{ir} + \tilde{F}_{vi})$. This process can be expressed mathematically as,

$$\text{Atten}_3(\tilde{F}_{ir} + \tilde{F}_{vi}) = \text{ReLU} \left( \text{Conv}_{1 \times 1} \left( \text{MAP}(\tilde{F}_{ir} + \tilde{F}_{vi}) \right) \right) \tag{9}$$

More specific details regarding the spatial domain module are depicted in Figure 5.

**Learning Strategy.** To enhance the training efficacy of the network, we employ mean square error (MSE) loss $L_{mse}$(Zhao et al., 2016), structural similarity index measure (SSIM) loss $L_{ssim}$, and L1 norms loss $L_{\ell_1}$ in the fusion scenes of rain and noise. In low-light and overexposed fusion scenes, we incorporate Exposure Control Loss $L_{exp}$ (Guo et al., 2020) to regulate the exposure level of the fusion result. For the first fusion scene, the total loss $L_{T1}$ can be expressed as follows:

$$L_{T1} = L_{mse} + L_{ssim} + L_{\ell_1} \tag{10}$$

Another fusion scene is represented as follows,

$$L_{T2} = L_{T1} + L_{exp} \tag{11}$$

By adjusting the loss of different tasks, the model can have better performance. Specific calculations on the image fusion loss can be found in (Yi et al., 2024; Huang et al., 2024)

## 4 EXPERIMENTS

To validate the effectiveness of the proposed algorithms, we conducted experiments on three types of interfering scenes: adverse weather (rain), noise (Gaussian noise), and exposure anomalies (low-light and overexposure). To ensure fairness in the experiment, our end-to-end framework is compared with existing "image restoration + fusion" combinations.

**Implementation details.** We trained separate models for different interference scenes. The proposed network was trained using the Adam optimizer with the initial learning rate set to $1e - 4$, gradually reduced to $1e - 6$ using cosine annealing strategy. The training process was carried out for 2000 epochs. To augment the training data, the input image undergoes random horizontal and vertical flips. The cropped image size during training was set to $128 \times 128$, and the batch size was 128. All experiments were conducted on a NVIDIA 3090 GPU using the PyTorch framework.

**Datasets.** For the training data: In the rain fusion scene, we randomly selected 1000 pairs of rain-containing images from the AWMM-100k dataset (Li et al., 2024b). In the noise fusion scene, we used 1000 pairs of images randomly selected from the $MSRS$ dataset (Tang et al., 2022b) and added Gaussian noise with a standard deviation of 10 to the visible images. In the overexposure scene, we scaled the pixel values in 1000 visible images from the $MSRS$ dataset to create overexposed images. In the low-light scene, we randomly selected 550 pairs of nighttime low-light images from the $MSRS$ dataset. For the test data: We randomly selected 50 images from the corresponding training dataset.

**Comparison Methods.** We selected seven state-of-the-art fusion methods for comparison: Co-CoNet (Liu et al., 2024a), Text-IF (Yi et al., 2024), CDDFuse (Zhao et al., 2023a), DeFusion (Liang et al., 2022), IGNet (Li et al., 2023d), LRRNet (Li et al., 2023b), and TGFuse (Rao et al., 2023). For image restoration, we incorporated Retinexformer (Cai et al., 2023) for low-light scene, Restormer (Zamir et al., 2022) for adverse weather and noise scenes, and MSEC (Afifi et al., 2021) for overexposed scene. We compared the combination of "restoration + fusion," similar to the approach used in Yi et al. (2024), which serves as a reasonable basis for comparison.

### 4.1 FUSION RESULTS IN COMPLEX SCENES

**Fusion Qualitative Comparison.** We conducted qualitative comparison experiments in four complex scenes: noise, rain, overexposure, and low-light. The results of all methods are shown in Figure 6. We incorporated an image restoration algorithm into each comparison method, resulting in fusion results from the combination of "restoration + fusion." Examination of the local zoomed-in areas in Figure 6 demonstrates that the proposed algorithm achieves superior fusion performance. It effectively removes noise and rain patterns from the source images, and when dealing with overexposure or low-light, it successfully restores image contrast and reconstructs detailed information.

**Fusion Quantitative Comparison.** For noise, rain, and overexposure scenes, we selected eight reference-based objective evaluation metrics: Normalized Mutual Information ($Q_{MI}$), Nonlinear Correlation Information Entropy ($Q_{NCIE}$), Image Fusion Metric Based on a Multiscale Scheme

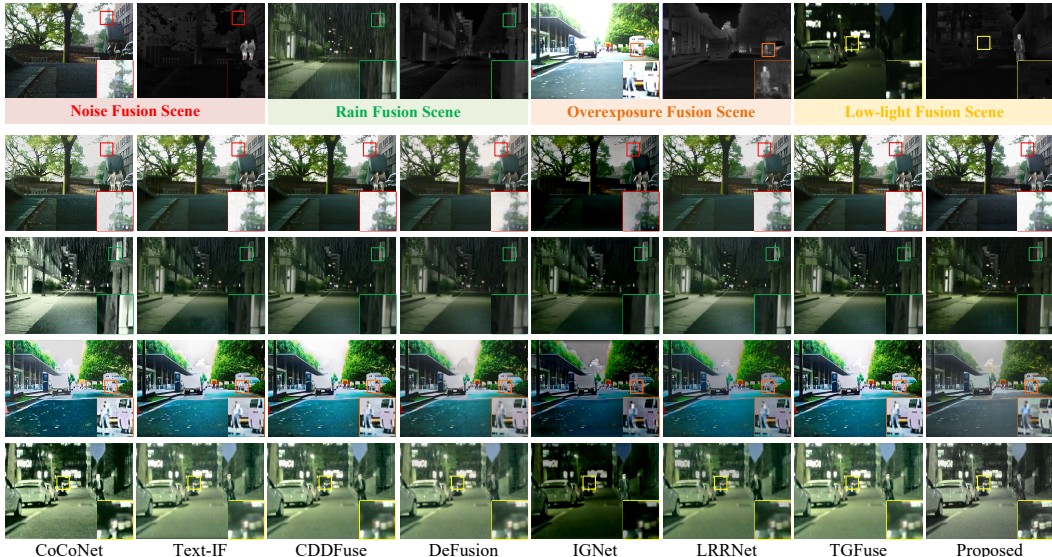

Figure 6: Qualitative comparison results on four complex scenes(noise, rain, overexposure and low-light). Image restoration models are added to each comparison methods.

Table 1: Non-reference-based metric results in noise, rain, overexposure and low-light scenes. **Bold** is the best and red is the second.

| Methods | Pub. | Noise | | Rain | | Overexposure | | Low-light | |
|---|---|---|---|---|---|---|---|---|---|
| | | AG | SF | AG | SF | AG | SF | AG | SF |
| CoCoNet (Liu et al., 2024a) | IJCV24 | **8.0572** | **23.0759** | **6.8417** | **19.2179** | **8.9615** | **29.8930** | **6.2875** | **16.5268** |
| Text-IF (Yi et al., 2024) | CVPR24 | 3.8155 | 11.1575 | 2.6657 | 7.7488 | 6.4451 | 20.9523 | 3.5552 | 9.1271 |
| CDDFuse (Zhao et al., 2023a) | CVPR23 | 3.8507 | 11.4911 | 2.5416 | 7.6948 | 6.4253 | 20.9130 | 3.5306 | 9.1545 |
| DeFusion (Liang et al., 2022) | ECCV22 | 2.7417 | 7.9073 | 1.8387 | 5.3740 | 4.8109 | 15.4282 | 2.9639 | 7.4264 |
| IGNet (Li et al., 2023d) | MM23 | 2.8226 | 7.9660 | 2.2003 | 5.9644 | 4.4915 | 14.0353 | 2.8894 | 8.2684 |
| LRRNet (Li et al., 2023b) | PAMI23 | 3.0373 | 9.2640 | 2.3891 | 7.4632 | 4.4625 | 14.3081 | 2.5653 | 6.6363 |
| TGFuse (Rao et al., 2023) | TIP23 | 3.7989 | 11.1444 | 2.5808 | 7.4661 | 6.4714 | 20.9396 | 3.5957 | 9.1830 |
| Proposed | – | 3.8916 | 11.5118 | 2.7703 | 9.3607 | 5.8244 | 15.1712 | 3.8320 | 12.9807 |

($Q_M$), Piella's Metric ($Q_S$), Chen-Blum Metric ($Q_{CB}$) (Liu et al., 2011), $SSIM$, Peak Signal-to-Noise Ratio ($PSNR$), and the Sum of the Correlations of Differences ($SCD$) (Aslantas & Bendes, 2015). For low-light scenes, since the test set was captured in real environments, we chose two non-reference-based objective evaluation metrics: average gradient ($AG$) and spatial frequency ($SF$) (Eskicioglu & Fisher, 1995). In addition, we conducted experiments on the evaluation of no-reference metrics in three other scenes. Higher values for all the aforementioned metrics indicate better image quality.

The quantitative comparison results are presented in Tables 1 and 2. In the noise, rain, and overexposure scenes, the proposed algorithms consistently rank in the top two across more than six metrics. In the no-reference metrics results, the CoCoNet achieved the highest scores in both metrics. However, as seen in Figure 6, the CoCoNet indiscriminately enhances pixel information, including interfering pixels, leading to high scores on no-reference evaluation metrics due to pixel redundancy. Quantitative comparisons across all four scenes demonstrate that the proposed algorithm has the best fusion performance in fusion tasks with interference.

**Segmentation Quantitative Comparison.** Here, we report the semantic segmentation accuracy of all methods in noisy scenes. In the segmentation task, we utilized the $MSRS$ dataset to conduct the training of the segmentation network (Peng et al., 2021). As shown in Table 3, our method achieves the highest mIoU score, indicating that the proposed algorithm outperforms the comparison methods in preserving the semantic information.

**Object Detection Quantitative Comparison.** In this section, we present the object detection accuracy of all methods in noisy scenes. The detection network (Wang et al., 2023) was trained using the $M3FD$ dataset. As shown in Table 6, our method achieved the highest AP@0.5 score, demon-

Table 2: Reference-based metric results in noise, rain, and overexposure scenes. **Bold** is the best and red is the second.

| Fusion | Pub. | Restoration | Noise | | | | | | | |
|---|---|---|---|---|---|---|---|---|---|---|
| | | | $Q_{MI}\uparrow$ | $Q_{NCIE}\uparrow$ | $Q_M\uparrow$ | $Q_S\uparrow$ | $Q_{CB}\uparrow$ | $SSIM\uparrow$ | $PSNR\uparrow$ | $SCD\uparrow$ |
| CoCoNet (Liu et al., 2024a) | IJCV24 | | 0.2695 | 0.8043 | 0.1584 | 0.4645 | 0.4388 | 0.1777 | 10.1291 | 1.4158 |
| Text-IF (Yi et al., 2024) | CVPR24 | | 0.4296 | 0.8079 | 0.3521 | 0.7779 | 0.4508 | 0.3066 | 14.6189 | 1.5290 |
| CDDFuse (Zhao et al., 2023a) | CVPR23 | | 0.4269 | 0.8076 | 0.3411 | 0.7596 | 0.4375 | 0.2543 | 14.8840 | 1.3573 |
| DeFusion (Liang et al., 2022) | ECCV22 | Restormer(Zamir et al., 2022) | 0.4331 | 0.8076 | 0.3368 | 0.7684 | 0.4496 | 0.2880 | 15.3989 | 1.1991 |
| IGNet (Li et al., 2023d) | MM23 | | 0.2908 | 0.8039 | 0.3562 | 0.5384 | 0.4226 | 0.2338 | 16.2459 | 1.5385 |
| LRRNet (Li et al., 2023b) | PAMI23 | | 0.4126 | 0.8068 | 0.2761 | 0.6975 | 0.4044 | 0.1081 | 16.0701 | 0.9270 |
| TGFuse (Rao et al., 2023) | TIP23 | | 0.3880 | 0.8067 | 0.3448 | 0.7743 | 0.4460 | 0.2942 | 14.4433 | 1.5248 |
| Proposed | – | w/o | **0.4504** | **0.8080** | **0.4072** | **0.8151** | **0.4609** | **0.3709** | 15.3349 | 1.4868 |
| | | | Rain | | | | | | | |
| CoCoNet (Liu et al., 2024a) | IJCV24 | | 0.2290 | 0.8036 | 0.2419 | 0.4011 | 0.4120 | 0.1486 | 9.1393 | 1.1987 |
| Text-IF (Yi et al., 2024) | CVPR24 | | 0.3250 | 0.8050 | 0.4785 | 0.7433 | 0.4178 | 0.2928 | 17.3649 | 1.2603 |
| CDDFuse (Zhao et al., 2023a) | CVPR23 | | 0.3372 | 0.8053 | 0.4560 | 0.7130 | 0.3929 | 0.2231 | 17.1871 | 1.1731 |
| DeFusion (Liang et al., 2022) | ECCV22 | Restormer(Zamir et al., 2022) | 0.3278 | 0.8049 | 0.3898 | 0.7337 | 0.4052 | 0.2550 | 18.1548 | 1.0192 |
| IGNet (Li et al., 2023d) | MM23 | | 0.2444 | 0.8031 | 0.3973 | 0.6766 | 0.4357 | 0.2706 | 19.3572 | 1.4356 |
| LRRNet (Li et al., 2023b) | PAMI23 | | 0.3254 | 0.8050 | 0.3755 | 0.6285 | 0.3736 | 0.0842 | 16.4855 | 0.7497 |
| TGFuse (Rao et al., 2023) | TIP23 | | 0.2883 | 0.8043 | 0.4708 | 0.7454 | 0.4232 | 0.2781 | 17.3183 | 1.2422 |
| Proposed | – | w/o | 0.2990 | 0.8040 | **0.4821** | **0.7857** | **0.4494** | **0.2963** | 19.0249 | 1.3798 |
| | | | Overexposure | | | | | | | |
| CoCoNet (Liu et al., 2024a) | IJCV24 | | 0.2378 | 0.8038 | 0.2010 | 0.5380 | 0.4226 | 0.2400 | 10.1644 | 1.2326 |
| Text-IF (Yi et al., 2024) | CVPR24 | | 0.4084 | 0.8079 | 0.3222 | 0.6908 | 0.4385 | 0.3505 | 11.5307 | 1.2650 |
| CDDFuse (Zhao et al., 2023a) | CVPR23 | | 0.4260 | 0.8082 | 0.3160 | 0.6915 | 0.4405 | 0.3442 | 11.3030 | 1.2495 |
| DeFusion (Liang et al., 2022) | ECCV22 | MSEC(Afifi et al., 2021) | 0.4132 | 0.8077 | 0.3297 | 0.7454 | 0.4631 | 0.3791 | 12.3898 | 1.1682 |
| IGNet (Li et al., 2023d) | MM23 | | 0.2984 | 0.8045 | 0.3279 | 0.5168 | 0.3941 | 0.2530 | 14.4742 | 1.4138 |
| LRRNet (Li et al., 2023b) | PAMI23 | | 0.4138 | 0.8076 | 0.3473 | 0.7209 | 0.4532 | 0.2979 | 13.1691 | 1.0324 |
| TGFuse (Rao et al., 2023) | TIP23 | | 0.3922 | 0.8077 | 0.3218 | 0.6990 | 0.4523 | 0.3497 | 11.3815 | 1.2331 |
| Proposed | – | w/o | **0.5444** | **0.8119** | **0.5143** | **0.8127** | **0.4964** | **0.4598** | 13.5599 | 1.3719 |

Table 3: Segmentation performance (mIoU) of all methods in noise scene. **Bold** is the best.

| Methods | Restoration | Background | Car | Person | Bike | Curve | Color Tone | Bump | mIoU |
|---|---|---|---|---|---|---|---|---|---|
| CoCoNet (Liu et al., 2024a) | | 98.18 | 86.22 | 68.20 | 69.13 | 56.70 | 60.02 | 61.45 | 71.41 |
| Text-IF (Yi et al., 2024) | | 98.51 | 89.61 | 73.00 | **70.63** | 63.70 | 63.41 | 77.45 | 76.62 |
| CDDFuse (Zhao et al., 2023a) | | 98.50 | 89.47 | 72.06 | 70.05 | **63.75** | 64.15 | 77.87 | 76.55 |
| DeFusion (Liang et al., 2022) | Restormer(Zamir et al., 2022) | 98.46 | 89.26 | 71.21 | 69.28 | 63.16 | **64.33** | **77.99** | 76.24 |
| IGNet (Li et al., 2023d) | | 98.33 | 88.31 | 72.60 | 68.39 | 54.75 | 61.93 | 71.29 | 73.66 |
| LRRNet (Li et al., 2023b) | | 98.17 | 87.71 | 64.58 | 66.08 | 49.99 | 61.55 | 77.76 | 72.26 |
| TGFuse (Rao et al., 2023) | | 98.49 | 89.33 | 73.17 | 69.78 | 62.18 | 63.79 | 77.85 | 76.37 |
| Proposed | w/o | **98.52** | **89.71** | **74.15** | 69.89 | 62.97 | 63.88 | 77.48 | **76.66** |

strating that the proposed algorithm outperforms the comparison methods in retaining significant target.

## 4.2 ABLATION EXPERIMENTS

**Impact of spatial domain.** We conducted ablation experiments by removing the spatial domain module. Processing solely in the frequency domain results in an inevitable loss of detail, which the spatial domain helps to recover. As shown in Table 5, the fusion performance consistently degrades across all four scenes when the spatial domain module is omitted.

**Impact of frequency guidance mechanism.** Utilizing the complementarity of multi-modal information, we design a multi-modal interactive guidance mechanism to facilitate the learning of crucial feature information in interference scenes. To validate this strategy, we conducted ablation experiments by removing the guidance mechanisms. As observed in Table 5, exchanging the infrared and visible mutual guidance mechanisms leads to a decline in fusion performance. Utilizing the amplitude affected by interference as the guiding image impedes the recovery of spatial information, resulting in lower scores across metrics. Additionally, removing the guidance mechanism hinders the interaction of information between different modalities, further degrading fusion performance.

## 4.3 COMPUTATIONAL EFFICIENCY ANALYSIS

We report the FLOPs, model size, and running time for all algorithms in Table 6. All experiments were conducted on source images of size 480×640. While CoCoNet and TGFuse have shorter running times, their models are larger. Although LRRNet has smaller FLOPs and parameters, its iterative approach to updating network parameters results in longer inference times. Furthermore, integrating different restoration algorithms for interference increases their computational complexity. Considering all three indicators, our method is the best overall.

Table 4: The detection accuracy of all methods in noise scene. **Bold** is the best.

| Methods | Restoration | People | Car | Bus | Lamp | Motorcycle | Truck | AP@0.5 |
|---|---|---|---|---|---|---|---|---|
| CoCoNet (Liu et al., 2024a) | | 0.791 | 0.889 | 0.889 | 0.646 | 0.641 | 0.762 | 0.770 |
| Text-IF (Yi et al., 2024) | | **0.816** | 0.836 | 0.872 | 0.785 | 0.625 | 0.723 | 0.776 |
| CDDFuse (Zhao et al., 2023a) | | 0.791 | 0.883 | 0.883 | 0.655 | 0.642 | 0.754 | 0.768 |
| DeFusion (Liang et al., 2022) | Restormer(Zamir et al., 2022) | 0.792 | 0.878 | 0.888 | 0.589 | 0.624 | 0.733 | 0.751 |
| IGNet (Li et al., 2023d) | | 0.793 | 0.859 | 0.836 | 0.532 | 0.554 | 0.721 | 0.716 |
| LRRNet (Li et al., 2023b) | | 0.764 | 0.89 | **0.89** | 0.7 | 0.653 | 0.755 | 0.775 |
| TGFuse (Rao et al., 2023) | | 0.791 | 0.889 | 0.889 | 0.684 | 0.635 | 0.762 | 0.775 |
| Proposed | w/o | 0.787 | **0.892** | 0.864 | **0.743** | **0.654** | **0.773** | **0.786** |

Table 5: Ablation experiment results in noise, rain and overexposure scenes. **Bold** is the best.

| Methods | Noise | | | | | | | |
|---|---|---|---|---|---|---|---|---|
| | $Q_{MI}\uparrow$ | $Q_{NCIE}\uparrow$ | $Q_M\uparrow$ | $Q_S\uparrow$ | $Q_{CB}\uparrow$ | $SSIM\uparrow$ | $PSNR\uparrow$ | $SCD\uparrow$ |
| w/o Spatial | 0.3906 | 0.8065 | 0.3452 | 0.7724 | 0.4486 | 0.3004 | **15.5512** | 1.4807 |
| Guidance Swap | 0.4469 | 0.8073 | 0.3514 | 0.8122 | 0.4549 | 0.3618 | 15.2575 | 1.4810 |
| w/o Guidance | 0.4501 | 0.8079 | 0.3993 | 0.8097 | 0.4506 | 0.3621 | 15.3800 | **1.5009** |
| Proposed | **0.4504** | **0.8080** | **0.4072** | **0.8151** | **0.4609** | **0.3709** | 15.3349 | 1.4868 |
| | Rain | | | | | | | |
| w/o Spatial | **0.3118** | 0.8038 | 0.4256 | 0.7862 | 0.4435 | 0.2879 | 18.5229 | 1.3649 |
| Guidance Swap | 0.9133 | 0.8037 | 0.4763 | 0.7828 | 0.4431 | 0.2948 | 18.8248 | 1.3780 |
| w/o Guidance | 0.3054 | **0.8041** | 0.4733 | **0.7859** | 0.4481 | 0.2920 | 18.7954 | 1.3716 |
| Proposed | 0.2990 | 0.8040 | **0.4821** | 0.7857 | **0.4494** | **0.2963** | **19.0249** | **1.3798** |
| | Over | | | | | | | |
| w/o Spatial | 0.5407 | 0.8107 | 0.4669 | 0.7949 | 0.4925 | 0.4424 | **14.3452** | 1.2024 |
| Guidance Swap | 0.5341 | 0.8112 | 0.5098 | 0.8114 | 0.4859 | 0.4590 | 13.6760 | 1.3570 |
| w/o Guidance | 0.5428 | 0.8117 | 0.5032 | 0.8102 | 0.4854 | 0.4583 | 13.5082 | 1.3298 |
| Proposed | **0.5444** | **0.8119** | **0.5143** | **0.8127** | **0.4964** | **0.4598** | 13.5599 | **1.3719** |

## 4.4 LIMITATION AND DISCUSSION

First, we validate the effectiveness of the proposed model in IVIF tasks. Future work will explore the application of the model in other multi-modality tasks such as medical image fusion. Second, we discuss the IVIF in four complex scenes, which should be extended to additional interference scenes in the future. More importantly, beyond improving fusion performance in complex scenes, this work examines the limitations of the "restoration + fusion" combination and provides a new idea for a unified model.

Furthermore, as a lightweight model with a limited number of parameters, achieving uniform weights for multiple scenes within a unified architecture presents significant challenges. The main advantage of a unified architecture lies in its consistent structure and adaptability across various contexts. This paper highlights how the proposed mutual bootstrapping mechanism effectively tackles challenges encountered in diverse complex scenes, em-

Table 6: The FLOPs, model size and running time (GPU-seconds for inference) of all methods.

| Methods | FLOPs(G) | SIZE(M) | TIME(ms) |
|---|---|---|---|
| CoCoNet (Liu et al., 2024a) | 10.39 | 9.12 | 23.3(1) |
| Text-IF (Yi et al., 2024) | 82.85 | 89.01 | 290.5 |
| CDDFuse (Zhao et al., 2023a) | 205.14 | 1.19(3) | 224.1 |
| DeFusion (Liang et al., 2022) | 3.82(1) | 7.87 | 49.9 |
| IGNet (Li et al., 2023d) | 16.49 | 7.87 | 32.6 |
| LRRNet (Li et al., 2023b) | 7.98(3) | 0.05(1) | 116.4 |
| TGFuse (Rao et al., 2023) | 3.99(2) | 137.34 | 23.4(2) |
| Proposed | 47.47 | 0.16(2) | 32.4(3) |

phasizing the importance of a unified architecture rather than uniform weights. We anticipate our work will inspire the development of more advanced models in the future.

## 4.5 CONCLUSION

We proposed a new perspective on addressing the IVIF problem in complex scenes based on the frequency domain. We conduct a thorough analysis of the degraded representation space of images in various complex scenes and propose a multi-modality information interaction guidance module. This module facilitates multi-modality feature interaction and extraction. Through extensive experiments conducted in four complex conditions: noise, rain, overexposure, and low-light, we demonstrate the effectiveness of the proposed algorithm in dealing with interfering information.

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

# A APPENDIX

## A.1 GENERALIZATION EXPERIMENTS IN MORE SCENES

In addition to the experiments conducted under interference scenes, we present the experimental results of various methods on the $M3FD$ and $LLVIP$ datasets in Tables 7 and 8. These datasets include numerous examples of ideal conditions or scenes with only slight visible light interference. The quantitative comparison experiments demonstrate that the proposed algorithm achieves the highest scores across five metrics on both the $M3FD$(Liu et al., 2022) and $LLVIP$(Jia et al., 2021) datasets. This finding indicates that our frequency-domain interaction guiding mechanism is effective not only in the complex scenes discussed earlier but also maintains excellent fusion performance in normal conditions, highlighting the strong generalization capability of the proposed algorithm. Furthermore, we present examples of the proposed algorithm on additional fused scenes in Figure 7. It can be observed that when the scene information provided by the infrared image is weaker than that of the visible image, meaning the infrared amplitude does not surpass the amplitude information of the visible image, the proposed interaction guidance mechanism still performs effectively. Proposed algorithm successfully retains the overall scene information from the visible image while capturing the significant thermal radiation information from the infrared image.

Table 7: Quantitative assessment results of $M3FD$ datasets. Maximum values are marked in blue.

| Methods | Pub. | $Q_{MI}\uparrow$ | $Q_{NCIE}\uparrow$ | $Q_M\uparrow$ | $Q_S\uparrow$ | $Q_{CB}\uparrow$ | $SSIM\uparrow$ | $PSNR\uparrow$ | $SCD\uparrow$ |
|---|---|---|---|---|---|---|---|---|---|
| CoCoNet (Liu et al., 2024a) | IJCV24 | 0.3013 | 0.8046 | 0.2071 | 0.5738 | 0.3737 | 0.4853 | 11.8118 | 1.6855 |
| Text-IF (Yi et al., 2024) | CVPR24 | 0.5279 | 0.8101 | **1.1252** | 0.8627 | 0.5343 | 0.7401 | 14.3888 | 1.4617 |
| CDDFuse (Zhao et al., 2023a) | CVPR23 | 0.5185 | 0.8101 | 0.5752 | 0.8435 | 0.5112 | 0.7249 | 13.4459 | 1.6421 |
| DeFusion (Liang et al., 2022) | ECCV22 | 0.3953 | 0.8055 | 0.3685 | 0.8204 | 0.4528 | 0.4264 | **16.1557** | 1.4204 |
| IGNet (Li et al., 2023d) | MM23 | 0.2858 | 0.8043 | 0.3975 | 0.6746 | 0.4241 | 0.6094 | 13.0875 | **1.7362** |
| LRRNet (Li et al., 2023b) | PAMI23 | 0.3765 | 0.8056 | 0.5250 | 0.8373 | 0.4869 | 0.7492 | 15.4582 | 1.6078 |
| TGFuse (Rao et al., 2023) | TIP23 | 0.5321 | 0.8106 | 0.6752 | 0.8464 | 0.5277 | 0.7202 | 13.2408 | 1.2955 |
| Proposed | _ | **0.5452** | **0.8145** | 0.5838 | **0.8676** | **0.6332** | **0.7529** | 13.9814 | 1.3271 |

Table 8: Quantitative assessment results of $LLVIP$ datasets. Maximum values are marked in blue.

| Methods | Pub. | $Q_{MI}\uparrow$ | $Q_{NCIE}\uparrow$ | $Q_M\uparrow$ | $Q_S\uparrow$ | $Q_{CB}\uparrow$ | $SSIM\uparrow$ | $PSNR\uparrow$ | $SCD\uparrow$ |
|---|---|---|---|---|---|---|---|---|---|
| CoCoNet (Liu et al., 2024a) | IJCV24 | 0.3206 | 0.8049 | 0.2226 | 0.6915 | 0.4544 | 0.3234 | 11.6255 | 1.7144 |
| Text-IF (Yi et al., 2024) | CVPR24 | 0.4448 | 0.8082 | 0.5221 | **0.8331** | **0.5223** | 0.4302 | 15.0227 | 1.6013 |
| CDDFuse (Zhao et al., 2023a) | CVPR23 | 0.6163 | 0.8159 | 0.5047 | 0.7552 | 0.4221 | **0.6435** | 14.6466 | 1.7061 |
| DeFusion (Liang et al., 2022) | ECCV22 | 0.4784 | 0.8095 | 0.2936 | 0.7978 | 0.4170 | 0.4147 | 15.7987 | 1.2455 |
| IGNet (Li et al., 2023d) | MM23 | 0.2943 | 0.8046 | 0.2704 | 0.6561 | 0.4208 | 0.5627 | 15.1792 | 1.4986 |
| LRRNet (Li et al., 2023b) | PAMI23 | 0.3623 | 0.8052 | 0.3046 | 0.7462 | 0.4276 | 0.6431 | 16.0880 | 1.0048 |
| TGFuse (Rao et al., 2023) | TIP23 | 0.6349 | 0.8148 | 0.2711 | 0.5360 | 0.4369 | 0.4335 | 13.0835 | 0.6546 |
| Proposed | _ | **0.6424** | **0.8176** | **0.5269** | 0.7861 | 0.4091 | 0.4489 | **16.4993** | **1.7869** |

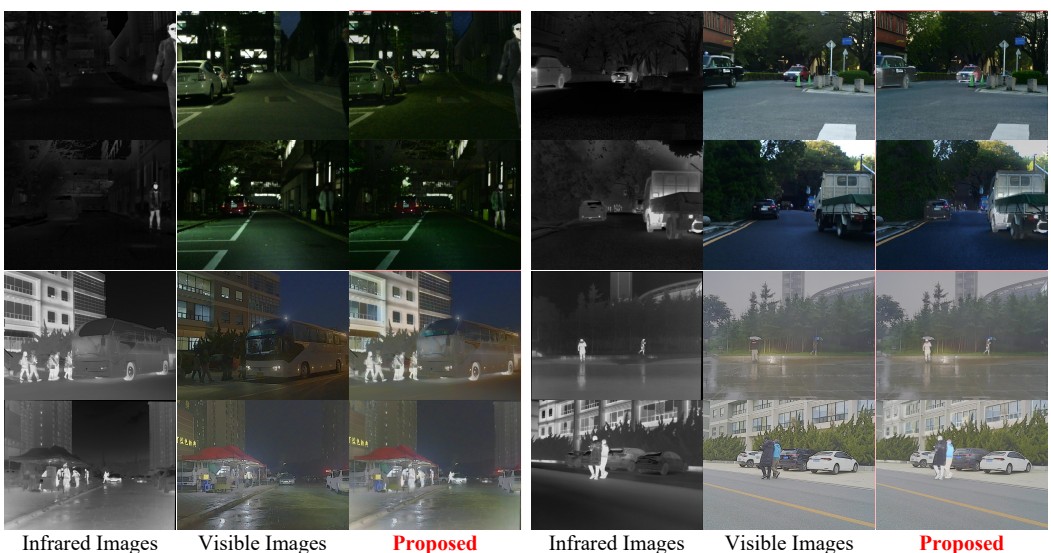

Figure 7: Visualization results on more fusion scenes.

