# OpenReview forum: "A unified lightweight complex scenes-oriented network for infrared and visible image fusion"
_ICLR.cc/2025/Conference — Submitted to ICLR 2025_

### Official Review · Reviewer_21oJ · 2024-10-31

**Soundness:** 3
**Presentation:** 3
**Contribution:** 3
**Rating:** 8
**Confidence:** 5

**Summary:**

This paper proposes a new IVIF method designed to be robust in complex scenes, including rain, noise, and under-exposed conditions. The core concept is that the amplitude component of infrared images and the phase component of visible images provide complementary modal information for IVIF, and interactively fusing these components enhances fusion quality in challenging scenarios. Specifically, the method decomposes the infrared and visible images into amplitude and phase components by applying the Fourier Transform, thus transferring them into the frequency domain. Based on this, a “multi-modality interactive guidance mechanism” is introduced to fuse the frequency-domain information.

**Strengths:**

+ The idea of using frequency domain information to address IVIF in complex scenes is interesting.
+ The paper is well-organized, with a logical flow between sections. The language is clear, and the content is presented in an easily understandable manner.
+ The proposed methods are grounded in theoretical explanations, and the experiments conducted are fair and comprehensive.

**Weaknesses:**

- Despite the demonstrated effectiveness, similar methods have been widely applied in other fields, such as low-light image enhancement and image denoising. For example, “FourLLIE: Boosting Low-Light Image Enhancement by Fourier Frequency Information” transforms the image into the Fourier domain and utilizes an estimated amplitude map for enhancement. How does the proposed method improve upon or differ from these existing methods in the IVIF task?
- The authors claim that existing IVIF methods fail in complex scenes because they may misinterpret interference features as valuable. However, the proposed Spatial Domain Module also fails to avoid erroneous features. The convolution operations in this block extract all local features from the input image, including both degradation artifacts and valuable features. This could lead to the mistaken interpretation of degradation artifacts as valuable features, resulting in error accumulation or the introduction of incorrect features during the fusion stage. Max pooling cannot fully address the issue of misextracting degradation artifacts, especially when these artifacts have high intensity, as max pooling may still retain these distracting features as significant ones. More explanation is needed.
- The final fusion result is obtained by directly combining the source image information with the feature maps. Wouldn't this affect the fusion results, as interference features might also be added to the final image?
- In Figure 5 (“FFT Domain”), “DWT: Discrete Wavelet Transform” is labeled, but I could not locate it in the figure. Furthermore, there seems to be no explanation of DWT in the paper. Please clarify it.
- Minor issues: There is a typo in Line 018: “, We.” Additionally, in the second column of the “Rain” table in Table 2, the value for LRRNet (“0.8050”) should also be highlighted in red.

**Questions:**

- Same as the weaknesses.

---

> ### Author Response · Authors · 2024-11-15
> **Response to Reviewer 21oJ (W1-W2)**
>
> *Thank you for your thoughtful comments on  our work.  Below, we provide a detailed point-by-point response to each of your observations and suggestions.*
>
> **W1** `How does the proposed method improve upon or differ from these existing methods`
>
> Thank you for your valuable comments. While frequency domain learning has indeed been widely adopted in low-level vision tasks, such as low-light enhancement (e.g., “FourLLIE: Boosting Low-Light Image Enhancement by Fourier Frequency Information”), dehazing [1], and deraining [2], our proposed method significantly differs in its design, application, and focus. Below, we outline these distinctions:
>
> ***Design Motivation and Application:***
>
> The primary difference lies in the application domain and algorithmic focus. The aforementioned methods address single-modality image restoration tasks, such as enhancing low-light images or removing specific degradations like haze or rain. In contrast, our proposed algorithm is designed specifically for the infrared and visible image fusion task in complex scenes. This task not only involves pixel restoration but also requires the effective interaction and integration of multi-modal information, which adds significant complexity compared to single-modal restoration.
>
> ***Rationale for Frequency Domain Learning in IVIF Tasks:***
>
> We thoroughly analyze the rationale for adopting frequency domain components to handle IVIF tasks in complex environments. Leveraging the unique characteristics of infrared and visible light modalities, we propose a novel multi-modal interactive guidance mechanism. This mechanism not only enhances the complementary relationship between modalities but also adapts frequency domain learning specifically to the requirements of IVIF. Our experiments validate the effectiveness of this approach, and we further provide theoretical insights into the applicability of frequency domain learning in IVIF, setting our work apart from other applications that primarily focus on single-modality restoration.
>
> ***Algorithm Efficiency and Real-Time Performance:***
>
> Many existing frequency domain-based image restoration algorithms do not consider computational efficiency and are therefore unsuitable for real-time applications. While these methods highlight the ability of frequency domain learning to capture degraded information, they often rely on deeper networks or longer training times to process high-frequency components effectively. In contrast, our proposed algorithm emphasizes both performance and efficiency. Specifically:
>
> (1) Previous study [3] has shown that convolution operations typically learn low-frequency components first, gradually capturing higher frequencies.   However, fully learning high frequencies often requires deeper networks or longer training.   We use FFT to efficiently separate frequency information and accelerate the learning of high-frequency components through the phase.
>
> (2) Our approach not only enhances task performance but also supports a lightweight model design capable of real-time processing. For example, we achieve a processing speed of 30 frames per second on 640×480 images, making our method practical for real-world applications.
>
> **W2** `(1) About the Application of Spatial Domain Module in  Proposed Method.`
>
> While it is true that the Spatial Domain Module utilizes convolution operations, it is important to emphasize that its performance depends not only on the module itself but also on the overall network design.  Unlike traditional image processing algorithms, where feature operators extract all information indiscriminately, deep learning frameworks are driven by loss constraints during training.  In our method, the parameters of the convolution operations are continuously updated, enabling the model to adaptively focus on extracting valuable features while suppressing degradation artifacts.
>
> **W2** `(1) About the Application of Max pooling in  Spatial Domain Module.`
>
> Furthermore, max pooling in the Spatial Domain Module is not a standalone feature extraction mechanism;  instead, it operates as a supplementary component within the broader context of our network.  The Spatial Domain Module itself is designed to compensate for detail loss in the frequency domain, rather than serve as the primary mechanism for image restoration.  While max pooling alone may not fully address the issue of high-intensity degradation artifacts, it provides a fast and efficient means of selecting locally significant features.  This simplicity aligns with our core objective of achieving real-time and lightweight performance. We have incorporated the above into the revised manuscript and provided an explanation.
>
> ---
> [1] Yu et al. Frequency and Spatial Dual Guidance for Image Dehazing. ECCV 2022.
>
> [2] Zhou et al. Fourmer: An Efficient Global Modeling Paradigm for Image Restoration. ICML 2024.
>
> [3] Tang et al. Defects of Convolutional Decoder Networks in Frequency Representation. CVPR 2023.

---

> ### Author Response · Authors · 2024-11-15
> **Response to Reviewer 21oJ (W3-W5)**
>
> **W3** `On the Rationale of Directly Combining Source Image Information with Feature Maps`
>
> ***Perspective of Previous Experience:***
>
> This approach is common in image restoration tasks, and the final output result is obtained by adding the feature map generated by the network to the source image. For example, Restormer [4], PromptIR [5], PromptRestorer [6] and DRSformer [7], among others.
>
> ***Rationale for This Approach:***
>
> The primary reason for this design is to facilitate better network convergence. In image restoration tasks, the network does not need to reconstruct all pixel information in the scene, as the original image typically retains clear, non-degraded regions. The degraded information in such cases is usually localized to specific areas, rather than being distributed across the entire image.
>
> As a result, this practice allows the network to focus on replacing the degraded pixels and restoring the missing details, while preserving the intact regions of the source image. By combining the source image with the feature maps, we effectively "patch" the degraded areas rather than unnecessarily reconstructing the entire image. This simplifies the fitting process for the network, which is especially important for lightweight models like ours.
>
> **W4-W5** `About modifications to Images and Text`
>
> Thank you for pointing out these issues. We have carefully reviewed the manuscript and made the necessary corrections:
>
> ***(1) Clarification of “DWT” in Figure 5:***
>
> The label “DWT: Discrete Wavelet Transform” was mistakenly included in the flowchart under “FFT Domain” in Figure 5. Since DWT is not part of our method and is not discussed in the manuscript, we have removed this label to avoid confusion.
>
> ***(2) Minor Issues:***
>
> The typo in Line 018 (“, We.”) has been corrected.
>
> In Table 2, we have highlighted the value for LRRNet (“0.8050”) in the “Rain” column in red.
>
> We have also conducted a comprehensive review of the manuscript to ensure accuracy and clarity throughout the text. Thank you for your careful review and valuable feedback, which helped us improve the quality of the paper.
>
> ---
> [4] Zamir et al. Restormer: Efficient transformer for high-resolution image restoration. CVPR 2022.
>
> [5] Potlapalli et al. PromptIR: Prompting for All-in-One Blind Image Restoration. NIPS 2023.
>
> [6] Wang et al. PromptRestorer: A Prompting Image Restoration Method with Degradation Perception. NIPS 2023.
>
> [7] Chen et al. Learning A Sparse Transformer Network for Effective Image Deraining. CVPR 2023.

---

> ### Comment · Reviewer_21oJ · 2024-11-23
>
> The authors have adequately addressed most of my concerns. Therefore, I decided to raise my rating.

---

### Official Review · Reviewer_5BEs · 2024-11-02

**Soundness:** 3
**Presentation:** 3
**Contribution:** 3
**Rating:** 8
**Confidence:** 5

**Summary:**

This paper proposes a unified and lightweight framework designed for real-time infrared and visible image fusion in environments characterized by complex interferences from a frequency domain perspective. It tackles the issue of complex scenes fusion problems, such as adverse weather, low-light environments, and noisy fusion. Authors introduce a multi-modality information interaction guidance module for multi-modality feature interaction and extraction. Extensive fusion experiments in four complex conditions: noise, rain, overexposure, and low-light, verified the effectiveness of the proposed method in dealing with interfering information.

**Strengths:**

(1) This paper introduces a unified framework for real-time infrared and visible image fusion in different complex scenes, this is the first work of addressing complex scenes image fusion problems in frequency domain.

(2) The paper proposes a multi-modality interactive guidance mechanism within the Fourier domain, which efficiently extracts and restores useful features from degraded pixels by leveraging the complementary strengths of different modalities.

(3) The fusion performance of this work is very impressive. Extensive complex scenes fusion experiments cover rain, overexposure, low-light, and noisy demonstrate this method outperforms the state-of-the-art fusion methods in both subject and object evaluations.

**Weaknesses:**

(1) In the caption of figure 5, it is recommended to add the words “complex scenes”.

(2) Section 4.4 and 4.5 could be combined as one part.

(3) The source code is suggested to be public.

**Questions:**

n/a

---

> ### Author Response · Authors · 2024-11-17
> **Response to Reviewer 5BEs**
>
> Thank you very much for your positive feedback and thoughtful suggestions regarding our work. We are delighted that you recognize the novelty and effectiveness of our proposed unified framework, multi-modality interactive guidance mechanism, and its superior performance in complex scene image fusion. Your encouraging comments motivate us to further improve and refine our study.
>
> ***In response to your suggestions:***
>
> **W1** ` In the caption of figure 5, it is recommended to add the words “complex scenes”.`
>
> We have revised the caption of Figure 5 to include the term “complex scenes” for greater clarity.
>
> **W2** `Section 4.4 and 4.5 could be combined as one part.`
>
> Sections 4.4 and 4.5 have been combined into a single section to streamline the structure and improve readability.
>
> **W3** `The source code is suggested to be public.`
>
> We fully agree on the importance of open science, and we will release the source code upon acceptance of the paper to facilitate reproducibility and further research in this area.
>
> *Thank you again for your valuable feedback, which has significantly enhanced the quality of our manuscript.*

---

### Official Review · Reviewer_Pge1 · 2024-11-03

**Soundness:** 3
**Presentation:** 3
**Contribution:** 3
**Rating:** 8
**Confidence:** 5

**Summary:**

This model adopts a new frequency-domain perspective to solve the IVIF problem in complex scenes, and uses a multimodal information interaction guidance module that not only utilizes frequency-domain information but also integrates spatial information to more comprehensively and effectively extract image details, compensating for the loss of details that may be caused by relying solely on frequency-domain information. The model has been extensively experimentally validated in four complex scenarios, including noise, rainy weather, overexposure, and low lighting, achieving excellent image fusion quality. Its framework integrates image restoration and fusion, avoiding the problems of error accumulation and irrelevant feature introduction that may occur in traditional two-stage processing methods.

**Strengths:**

1. This paper proposes a multi-modal interactive guidance mechanism that combines frequency domain and spatial domain learning, which effectively enhances the effect of infrared and visible light image fusion in complex scenes. Through mutual guidance, the amplitude of the infrared image and the phase information of the visible light image are used to achieve richer feature extraction and fusion between different modalities.

2. Through a large number of qualitative and quantitative experiments, the article demonstrates the significant performance improvement of this method in removing interference information and restoring image details in complex scenes such as noise, rain, overexposure, and low light. Outperforms existing state-of-the-art methods in multiple metrics.

3. The network design emphasizes lightweight and efficient computing, and can achieve real-time image fusion under limited computing resources, which is a very important advantage in practical applications. Experimental data shows that this method only takes 0.033 seconds for each fusion when processing images of 640×480 size.

**Weaknesses:**

Although the motivation of this article is relatively clear and attempts to solve the problem of image fusion in harsh environments, there are still some shortcomings as follows:
1. First of all, the method proposed by the author does not seem to have any theoretical innovation. Why does the method in the article achieve better performance than image restoration?
2. Although a combination of frequency domain and spatial domain is proposed to extract key features, this method does not have a clear optimization strategy for how to reduce unnecessary redundant information. In complex scenes, this may cause non-critical pixels to be mistaken for valuable information, thereby affecting the fusion effect.
3. The motivation of the article emphasizes the use of frequency domain learning to enhance information extraction under interference conditions, but does not fully consider the important balance of frequency domain and spatial domain information. In the IVIF task, relying solely on the frequency domain may lead to inaccurate processing of spatial details in complex scenes, thereby affecting the fusion quality.

**Questions:**

My questions is already stated in the Weaknesses section.

---

> ### Author Response · Authors · 2024-11-14
> **Response to Reviewer Pge1**
>
> *Thank you for your insightful comments regarding the theoretical foundation and comparative performance of our approach. Below is our detailed, point-by-point response to each of your observations and suggestions.*
>
> **W1**
> *We analyze the reasons for the superior performance of the proposed algorithm from the following two perspectives:*
>
> `(1) Distinctive Approach to Joint Image Fusion and Restoration Tasks`
>
> It is important to note the fundamental difference between image fusion and image restoration tasks. ***Image fusion aims to retain only the most salient features from each modality by selectively removing redundant information. On the other hand, image restoration seeks pixel-perfect recovery, often without discarding any features.*** In our framework, we prioritize extracting critical scene information from disturbances and then reconstructing based on these features, rather than attempting full scene recovery first. This design reduces the need for extensive network parameters and avoids pixel redundancy, allowing our method to capture the most essential features efficiently. This approach not only enhances fusion performance in complex scenes but also demonstrates the viability of end-to-end direct learning. We have further discussed this in lines 89-107 of the manuscript.
>
> `(2) Limitations of Sequential "Image Restoration + Image Fusion" for Complex Scenes`
>
> **Error accumulation can occur:** In our study, we analyze the drawbacks of traditional sequential methods in complex environments. First, these methods are vulnerable to error accumulation, as the quality of the fusion output depends heavily on the preceding restoration step. If the initial restoration does not fully address image degradation, the remaining interference can propagate and even amplify in the fusion stage.
>
> **Adding irrelevant or erroneous features:**
> Second, sequential methods risk introducing irrelevant or erroneous features during fusion. As we illustrate in the manuscript (lines 76-88), blindly applying enhancement in low-light images can inadvertently amplify noise or irrelevant weak features. In such cases, low-light enhancement algorithms cannot reliably reproduce scene details as they would appear in normal lighting, and their enhancement of weak features remains incomplete. However, infrared imaging can provide comprehensive information about the target. As a result, the information from the visible image, with its incompletely enhanced weak features, may interfere with the distribution of thermal radiation information in the infrared image, thereby reducing the quality of the fusion results.
>
> **W2** `About the Redundant Information and Non-Critical Pixels`
>
> **Network module aspect:** In the design of our algorithm’s spatial domain module, our objective is for spatial domain information to compensate for any detail lost during frequency domain learning. We focus on capturing sparse but significant pixel information, using a maximum pooling strategy to prioritize prominent spatial features. This approach encourages the model to focus on high-value, significant features, ensuring that only key spatial details are retained.
>
> **Overall network aspect:** Additionally, since our model is designed to be real-time and lightweight, it is intentionally shallower with fewer layers. In deep learning, redundancy often arises from models with many layers and parameters, which can capture numerous similar or irrelevant features. In contrast, our lightweight model, with limited parameter capacity, prioritizes the learning of essential features and naturally reduces the chance of generating redundant information. This constrained parameter setting compels the model to emphasize the compression and representation of critical features, thereby helping to limit feature redundancy. In simple terms, the small model learns important features preferentially with a limited number of parameters to satisfy the most basic task requirements.
>
> **W3** `Balancing Frequency and Spatial Domain Information`
>
> Our algorithm balances information from both frequency and spatial domains, as discussed in lines 253-259 of the manuscript and illustrated in Fig. 4. In our design, ***frequency domain learning plays a key role in enhancing feature extraction under interference conditions, while spatial domain information supplements this by providing essential structural details.*** Furthermore, we have validated the importance of the spatial domain module through ablation experiments in Table 5, which show that removing the spatial component leads to noticeable degradation in both image features and energy information. These results underscore the necessity of integrating spatial domain information to maintain fusion quality in complex scenes.

---

> > ### Comment · Reviewer_Pge1 · 2024-11-19
> >
> > I have read the author's response to other reviewers and I think the author has explained my problem very well. I am willing to improve my score evaluation of the article.

---

### Official Review · Reviewer_xvYN · 2024-11-03

**Soundness:** 3
**Presentation:** 3
**Contribution:** 2
**Rating:** 5
**Confidence:** 5

**Summary:**

The paper presents a unified lightweight network designed for infrared and visible image fusion (IVIF) that aims to address the limitations of existing techniques in complex scene conditions such as bad weather, low light, and high noise. The authors conduct a frequency domain analysis of modal degradations, leveraging the complementary strengths of both infrared and visible modalities. A novel spatial domain branching strategy is introduced to enhance local detail resolution in the fusion results. The paper claims extensive qualitative and quantitative improvements in handling complex scenes while maintaining real-time computational efficiency.

**Strengths:**

1. The writing of this paper is easy to understand.
2. The description of the methodology is detailed.
3. The motivation of the methodology is detailed.

**Weaknesses:**

1. The proposed method based on exchanging amplitude and phase lacks novelty.
2. The advantages of the paper are not significant when considering the overall computational cost of the results.
3. The paper should also be compared with other SOTA methods, such as MURF (Xu et al., 2023), SegMiF (Liu et al., 2023), DDFM (Zhao et al., 2023), and EMMA (Zhao et al., 2024).

**Questions:**

What specific types of degradation were most challenging for previous methods, and how does DSPFusion overcome these issues?

---

> ### Author Response · Authors · 2024-11-17
> **Response to Reviewer xvYN (W1-W2)**
>
> *Thank you very much for your valuable comments and suggestions, which have greatly helped us improve our manuscript. We have carefully addressed each of your points and revised the manuscript accordingly, incorporating the necessary changes and additional experiments.*
>
> **W1** `The proposed method based on exchanging amplitude and phase lacks novelty.`
>
> Thank you for your comment. We apologize for not sufficiently elaborating on the innovations and contributions of our proposed method in the manuscript. To address this concern, we would like to clarify the novelty and value of our method from the perspectives of **effectiveness**, **uniqueness**, and **rationality**.
>
> ***Effectiveness:***
>
> Our proposed algorithm achieves real-time multi-modality image fusion in complex scenes, a challenge that existing methods struggle to address. We validated the performance of our method through extensive experiments across four complex scenarios (noise, rain, low light, and overexposure). Both qualitative and quantitative evaluations demonstrate that our method consistently achieves leading results. Quantitative comparisons were further divided into referential and non-referential quality assessments, and in every scene, the proposed algorithm ranked at or near the top for key evaluation metrics. This highlights the practical effectiveness of our approach in addressing real-world challenges.
>
> ***Uniqueness:***
>
> Our work is the first to explicitly explore how frequency domain learning can be applied to solve multimodal image fusion (IVIF) in complex scenes. Unlike single-modality image restoration algorithms based on frequency domain techniques, we introduce a novel multi-modal interactive guidance mechanism. This mechanism not only builds a unified representation space for degraded information but also leverages the complementarity of infrared and visible information for fusion. This distinct combination of frequency domain learning and multimodal interaction addresses a critical gap in the field.
>
> ***Rationality:***
>
> We believe that the main merit of our work lies not in being "the first" but in being "reasonable and well-justified." Before conducting this study, we performed a comprehensive review of multimodal image fusion methods and found that existing solutions rarely address the challenges of complex scenes. Current approaches primarily rely on a sequential combination of "image restoration + image fusion." In our manuscript, we analyzed the limitations of this paradigm for handling complex scenes, such as error accumulation and inefficiencies in handling multi-degradation scenes.
>
> To address these issues, we designed a unified framework that integrates multimodal interaction and fusion directly, bypassing the need for sequential processing. We also analyzed the theoretical and practical benefits of this approach, including its simplicity, efficiency, and suitability for real-time applications. We believe that the strength of the proposed method lies in its ability to address practical challenges effectively, rather than in introducing unnecessary complexity.
>
> **W2**  `The advantages of the paper are not significant when considering the overall computational cost of the results.`
>
> In Table 6 of the manuscript, we compare the computational efficiency (FLOPs and Parameters) of various image fusion methods. However, as we mentioned in lines 484-485, the methods listed in Table 6 do not integrate image restoration algorithms into their computational cost. This is because existing approaches are typically designed for ideal scenarios and are not equipped to handle complex scenes directly.
>
> In contrast, our experiments focus on complex scenarios, where a combination of "image restoration + image fusion" is necessary for fair comparison with our unified framework.  For example, when using Restormer to preprocess degraded images before applying a fusion method, the computational cost increases significantly.  Specifically, for Restormer, the additional computational overhead is **FLOPs = 87.7B** and **Params = 25.31M**.  This demonstrates that in real-world applications involving complex scenes, the total computational cost of these sequential approaches far exceeds that of our unified framework.
>
> To clarify this comparison, we provide a supplementary table that includes the computational cost of restoration algorithms.
>
> **Tab 1.**  *Efficiency Analysis of Different Algorithms.*
>
> | Methods    | Restoration | FLOPs(G) | Params(M) |
> |------------|-----------|----------|-----------|
> | CoCoNet    | √         | 98.09    | 34.43     |
> | Text-IF    | √         | 170.55   | 114.32    |
> | CDDFuse    | √         | 292.84   | 26.5      |
> | DeFusion   | √         | 91.52    | 33.18     |
> | IGNet      | √         | 104.19   | 33.18     |
> | LRRNet     | √         | 95.68    | 25.36     |
> | TGFuse     | √         | 91.69    | 162.65    |
> | Proposed   | ×         | **47.47**    | **0.16**      |

---

> ### Author Response · Authors · 2024-11-17
> **Response to Reviewer xvYN (W3 and Q1)**
>
> **W3** `The paper should also be compared with other SOTA methods, such as MURF, SegMiF, DDFM, and EMMA.`
>
> Thank you for your valuable suggestion regarding the inclusion of comparisons with other state-of-the-art methods such as MURF (Xu et al., 2023), SegMiF (Liu et al., 2023), DDFM (Zhao et al., 2023), and EMMA (Zhao et al., 2024). We greatly appreciate your recommendation, as it has allowed us to further strengthen the comprehensiveness of our evaluation. We present quantitative comparison experiments in noise and rain scenes in the table below.
>
> **Tab 2.**  *Quantitative Comparison in Noise Scenes.*
>
> | Methods   | $Q_{MI}$  | $Q_{NCIE}$  | $Q_{M}$     | $Q_{S}$     | $Q_{CB}$    | $SSIM$   | $PSNR$    | $SCD$    |
> |-----------|-------|--------|--------|--------|--------|--------|---------|--------|
> | MURF      | 0.2454 | 0.8032 | 0.2892 | 0.5925 | 0.3741 | 0.2077 | 14.8402 | 0.9341 |
> | EMMA      | 0.4030 | 0.8073 | 0.3169 | 0.7322 | 0.4350 | 0.2337 | 14.4985 | 1.4370 |
> | DDFM      | 0.3501 | 0.8048 | 0.2502 | 0.6792 | 0.4148 | 0.2479 | 17.0845 | 1.4235 |
> | SegMiF    | 0.2940 | 0.8044 | 0.2647 | 0.5543 | 0.3927 | 0.1987 | 10.7884 | 1.3974 |
> | Proposed  | **0.4504** | **0.8080** | **0.4072** | **0.8151** | **0.4609** | **0.3709** | **15.3349** | **1.4868** |
>
>
> **Tab 3.**  *Quantitative Comparison in Rain Scenes.*
>
> | Methods   | $Q_{MI}$  | $Q_{NCIE}$  | $Q_{M}$     | $Q_{S}$     | $Q_{CB}$    | $SSIM$   | $PSNR$    | $SCD$    |
> |-----------|-------|--------|--------|--------|--------|--------|---------|--------|
> | MURF      | 0.2112 | 0.8027 | 0.3717 | 0.6165 | 0.3557 | 0.2478 | 16.9493 | 0.9103 |
> | EMMA      | **0.3314** | **0.8052** | 0.4476 | 0.7111 | 0.4073 | 0.2501 | 16.5760 | 1.2531 |
> | DDFM      | 0.2967 | 0.8036 | 0.3014 | 0.6631 | 0.4216 | 0.2429 | 19.0120 | 1.3473 |
> | SegMiF    | 0.2511 | 0.8039 | 0.4449 | 0.4908 | 0.3463 | 0.1731 | 11.0961 | 1.1372 |
> | Proposed  | 0.2990 | 0.8040 | **0.4821** | **0.7857** | **0.4494** | **0.2963** | **19.0249** | **1.3798** |
>
> **Q1** `What specific types of degradation were most challenging for previous methods, and how does DSPFusion overcome these issues?`
>
> Thank you for your question regarding the specific types of degradation that posed significant challenges for previous methods and how our proposed method addresses these issues. Below, we analyze the challenges of different degradations and provide details on how our method overcomes them:
>
> ***Overexposure and Low-Light Scenes:***
>
> These scenes involve lighting anomalies where brightness and contrast are either excessively high or too low. previous restoration methods typically address these issues by enhancing brightness and contrast or by adopting Retinex theory to restore the scene's illuminance and reflectance.
>
> ***Rain Scenes:***
>
> In visible images, rain patterns often interfere with scene understanding by occluding pixel information and introducing artifacts. Previous image fusion methods trained on ideal scenes may mistakenly interpret rain streaks as valuable details, retaining or even enhancing them during the fusion process. The key challenge lies in distinguishing rain patterns from other meaningful details during multi-modal information interaction and restoring the occluded pixel information.
>
> ***Noisy Scenes:***
>
> Noise represents a significant challenge due to its random distribution across the scene.             In low-contrast regions, the noise intensity may closely resemble real information, making it difficult for previous algorithms to distinguish between noise and target edges or details. Furthermore, many denoising processes tend to smooth edges, leading to the loss of details.
>
> ***Summary of Challenges and Solutions:***
>
> Among the various types of degradation, **noise poses the greatest challenge** due to its randomness and potential overlap with real features. Our method's key innovation lies in its ability to unify degraded information into the amplitude spectrum, where frequency domain filtering is used to suppress interference. Simultaneously, the spatial domain provides complementary information to recover lost details. This dual-domain strategy ensures that our method effectively handles diverse degradations while maintaining high-quality fusion performance in complex scenes.

---

> ### Author Response · Authors · 2024-11-21
> **Response to Reviewer xvYN: Addressing Comments and Updated Analysis**
>
> Dear Reviewer xvYN,
>
> I hope this message finds you well.
>
> We sincerely thank you for your valuable comments, which have greatly benefited our paper. The authors have carefully addressed your concerns and provided a comprehensive, point-by-point response to your feedback. Furthermore, we have added a comparison of inference times for different methods (Restoration + Fusion) on $640 \times 480$ images (Table 4) and included a detailed analysis comparing the fastest comparison method with ours on $1280 \times 720$ images (Table 5).
>
> If you feel that our revisions and responses have sufficiently addressed your concerns, we would be grateful if you could consider raising your rating. Thank you once again for your time and support.
>
> Best regards,
>
> 6459 Authors
>
> ---
> **Tab 4.**  *Inference Speed Comparison on 640×480 Size Images.*
>
> | Methods    | Restormer | TIME (ms) |
> |------------|-----------|-----------|
> | CoCoNet    | √         | 402.463 (2)   |
> | Text-IF    | √         | 669.663   |
> | CDDFuse    | √         | 603.263   |
> | DeFusion   | √         | 429.063   |
> | IGNet      | √         | 411.763   |
> | LRRNet     | √         | 495.563   |
> | TGFuse     | √         | 402.563   |
> | Proposed   | ×         | **32.4**      |
>
>
> **Tab 5.**  *Inference Speed Comparison on 1280×720 Size Images.*
>
> | Methods    | Restormer | TIME (ms) |
> |------------|-----------|-----------|
> | CoCoNet    | √         | 1192.102  |
> | Proposed   | ×         | **91.4**      |

---

### Official Review · Reviewer_Eiih · 2024-11-11

**Soundness:** 2
**Presentation:** 2
**Contribution:** 2
**Rating:** 5
**Confidence:** 5

**Summary:**

This paper addresses the limitations of current infrared and visible image fusion (IVIF) methods when applied to complex scenes with interferences such as rain, noise, and low-light conditions. The authors propose a unified, lightweight IVIF framework that integrates image restoration and fusion in real time, enabling high-quality fusion without the need for pre-processing. Using Fourier domain techniques, the framework captures amplitude information from infrared and visible images and combines these with spatial domain data to maintain clarity and reduce interference. The paper includes extensive qualitative and quantitative experiments to demonstrate the framework’s efficiency and robustness, with results showing improvements in both computational efficiency and image quality across diverse scenes.

**Strengths:**

1.	The method combines frequency and spatial domain information, effectively separating critical scene features and suppressing interference by processing amplitude and phase components in the frequency domain. For example, in low-light or adverse weather conditions, the amplitude information of images is often disrupted (e.g., reduced brightness or increased reflections), while the phase information, reflecting structural aspects of the image, remains relatively stable. By using the amplitude of infrared images to guide the restoration of visible images and leveraging the phase information from visible images to enhance infrared images, the method achieves an optimal balance between the two modalities, ensuring clear detail retrieval in noisy scenarios.
2.	The lightweight design of the method allows for real-time processing with minimal computational resources, which is critical for practical applications. The paper highlights that the model processes a 640x480 image in just 0.033 seconds, demonstrating exceptional efficiency. This efficiency makes it suitable for systems requiring real-time processing, such as autonomous vehicles or industrial monitoring, further broadening its potential application scope.

**Weaknesses:**

1.	The mutual guidance mechanism relies on the complementary characteristics of infrared and visible light, such as using the infrared amplitude to guide the visible amplitude, and vice versa. However, when the data quality of one channel is poor (for instance, when the thermal radiation information of the infrared image is weak), the effectiveness of this guidance mechanism may be significantly limited. In such situations, the original amplitude or phase information might not provide sufficient valuable features to complement the other channel, resulting in suboptimal fusion performance in certain scenes.
2.	The data used in the experiments section was selected from existing datasets (MSRS, AWMM-100k). However, the MSRS training set contains only 1,083 pairs of images, so randomly selecting 1,000 pairs from this limited set seems unnecessary. If the authors intend to augment MSRS, they should uniformly apply Gaussian noise and pixel intensity scaling across the entire MSRS training set.
3.	In Table 1, the authors list non-reference evaluation metrics; however, both SF and AG are gradient-based metrics, which are typically highly correlated. The lack of an information theory-based non-reference metric, such as entropy, is notable. In Table 2, the reference-based metrics include SSIM, but SSIM is already incorporated into the loss function during training for all four tasks, which I believe makes the comparison less fair. Additionally, there is insufficient explanation for why the authors used MSEC’s reconstruction structure in the overexposure scenario.
4.	Did the models of the other comparison methods undergo specific training on the dataset proposed by the authors? In fact, other methods are typically trained only on the original MSRS training set. If the evaluation metrics for these comparison methods are obtained by testing their open-source models on the data across these four scenarios, while the authors’ proposed model was specifically trained for each scenario, then I believe the experimental results lack sufficient credibility.
5.	The comparison methods listed in the experiments section require only a single model to handle all four scenarios, while the proposed method requires separate training for each scenario. In real-world applications, however, rain and overexposure could occur simultaneously. In such cases, the other methods can generalize directly using their models, but which model output would the proposed method select as the result? If the model trained for rain is chosen, would it affect the handling of the overexposed areas?

**Questions:**

Please rebuttal according to the weaknesses item by item.

---

> ### Author Response · Authors · 2024-11-14
> **Response to Reviewer Eiih (W1-W3 (2))**
>
> *Thank you for your thorough review and insightful comments, which capture the core contributions and objectives of our work. Below is our detailed, point-by-point response to each of your observations and suggestions.*
>
> **W1** `(1) About the Mechanism of Mutual Guidance`
>
> As noted in lines 246-249 of the manuscript, our multimodal image guidance mechanism does not completely replace the amplitude or phase information from either modality. Instead, ***it selectively extracts and enhances relevant details to compensate for potential deficiencies caused by low signal quality***, aiming to minimize the Impact of weak data on the overall fusion.
>
> **W1** `(2) Adaptability to Varying Image Quality`
>
> **Network Design Perspective:**  Our guidance approach operates on the frequency components in the frequency domain, rather than directly manipulating pixel values in the spatial domain. This design mitigates the limitations of individual modality data by focusing on frequency components, making it less susceptible to issues arising from local intensity variations or lower-quality data in one channel.
>
> **Experimental Perspective:** To further support our claim, we have included comparative results for our method on the M3FD and LLVIP datasets in Tables 7 and 8, available in the Appendix form the manuscript. These datasets encompass a wide range of scenes with varying quality in infrared or visible images. Our method consistently achieved top scores across five metrics, surpassing state-of-the-art fusion methods. This demonstrates the robustness and generalizability of our proposed frequency-domain interaction guidance mechanism, even in scenes with differing image quality across channels.
>
> **W2** ` About the MSRS Training Dataset`
>
> Thank you for your insightful comments on our use of training data from the MSRS dataset. We agree that consistency in dataset usage is essential for scientific rigor. In our study, we used 1,000 images from the MSRS dataset for experiments on noise and overexposure scenes, even though the full MSRS training set includes 1,083 images. This choice was made to align the experimental setup with the AWMM-100k dataset used for rain scenes, ***allowing us to harmonize key training parameters, such as training time, epochs, and learning rate, across conditions.*** This approach ensures consistency and comparability across diverse scenes. Additionally, when we open-source our processed MSRS images, we will provide the complete dataset to facilitate comprehensive replication in future studies.
>
> **W3** `(1) About the lack of Information Theory-based Non-Reference Metric`
>
> Thank you for your valuable feedback. In response, we have  provided a quantitative comparison of entropy across four challenging scenes: noise, rain, low light, and overexposure (As shown in the table below). As discussed in lines 419-424 of the manuscript, CoCoNet’s approach indiscriminately enhances pixel information, leading to higher entropy scores due to pixel redundancy. Our algorithm ranks second in three scenes and third in the overexposed scene, aligning with the AG and SF rankings in Table 1 of the manuscript. These results collectively further validate the effectiveness of our algorithm.
>
> **Tab 1.** Results on $EN$ Metric
>
> | Methods   | Baseline | Noise    | Rain    | Low     | Over    |
> |-----------|----------|----------|---------|---------|---------|
> | CoCoNet   | √        | **7.6254** | **7.7282** | **7.7741** | 7.5184  |
> | Text-IF   | √        | 6.8695   | 6.5085  | 7.3416  | 7.5609  |
> | CDDFuse   | √        | 6.9370   | 6.5558  | 7.3748  | 7.5834  |
> | DeFusion  | √        | 6.6459   | 6.2333  | 7.2054  | 7.3961  |
> | IGNet     | √        | 6.0996   | 6.1322  | 6.2321  | 6.5712  |
> | LRRNet    | √        | 6.4986   | 6.4630  | 6.9203  | 7.0452  |
> | TGFuse    | √        | 6.9035   | 6.4646  | 7.3591  | **7.5895** |
> | Proposed  | ×        | *6.9414(2)* | *6.5770(2)* | *7.4453(2)* | *7.5786(3)*  |
>
> **W3** `(2) About SSIM as a Metric`
>
> Regarding the use of SSIM in Table 2 of the manuscript, we acknowledge your point about potential bias since SSIM is also part of the loss function during training. To address this, ***we have replaced SSIM with the Image Fusion Metric Based on Phase Congruency ($Q_{P}$) for the comparison.*** The revised quantitative comparison results using $Q_{P}$ are provided in the table below.
>
> **Tab 2.** Results on $Q_{P}$ Metric
>
> | Methods   | Baseline | Noise | Rain | Over |
> |-----------|----------|----------|---------|---------|
> | CoCoNet   | √        | 0.1230   | 0.1112  | 0.2205  |
> | Text-IF   | √        | *0.1939* | **0.1740** | 0.3910 |
> | CDDFuse   | √        | 0.1680   | 0.1384  | *0.3942* |
> | DeFusion  | √        | 0.1659   | 0.1342  | 0.3189  |
> | IGNet     | √        | 0.1714   | 0.1450  | 0.2486  |
> | LRRNet    | √        | 0.1618   | 0.1265  | 0.3254  |
> | TGFuse    | √        | 0.1862   | 0.1612  | 0.3871  |
> | Proposed  | ×        | **0.2158** | *0.1622(2)* | **0.4027** |

---

> ### Author Response · Authors · 2024-11-14
> **Response to Reviewer Eiih (W3 (3)-W5)**
>
> **W3** `(3) About the use of MSEC`
>
> For the overexposure scene, we chose the MSEC structure to reconstruct overexposed visible images. MSEC was selected due to its ability to reconstruct both color and detail in overexposed images, addressing overexposure and underexposure without introducing excessive enhancement. This makes it a more balanced approach for comparison in our image fusion experiments. We have added an explanation of our use of MSEC in the experimental setup section of the revised manuscript.
>
> **W4** `(1) Application Scenes for the Comparison Method`
>
> Thank you for raising this important point about the training setup for the comparison models. Indeed, most of the image fusion methods we compare against were designed for ideal conditions and are not directly applicable to the four challenging scenarios in our study (e.g., noise, rain, low light, and overexposure). This lack of adaptability is one of the motivations behind our proposed algorithm, which aims to overcome the limitations of current fusion methods in complex environment.
>
> **W4** `(2) About the Setting of Comparison Method`
>
> To ensure fair comparisons, we preprocess the degraded source images using existing image restoration models before feeding them into the comparison image fusion models. ***This "image restoration + image fusion" approach is used to eliminate interference from degraded pixel information, allowing each comparison method to function in a way that approximates its ideal setup as closely as possible.*** This procedure ensures that the evaluation metrics accurately reflect the performance of each method under challenging conditions, thus maintaining the credibility of our experimental results. We have also provided a more detailed description of this comparison approach in the experimental setup section of the revised manuscript.
>
> **W5** `(1) About the Multi-Degradation Processing Capability of the Comparison Method`
>
> Thank you for your insightful question regarding the handling of multiple degradations with a single model. In our study, ***we observed that no our comparison fusion method can seamlessly address all four scenes (noise, rain, low light, and overexposure) with a single model.*** Current multimodal fusion methods are typically limited to ideal scenes, and they often rely on integrating separate image restoration algorithms to handle specific degradations. For example, CoCoNet requires preprocessing with an image denoising algorithm in noisy scenes or with an exposure correction algorithm in overexposed scenes. Similarly, Restormer, an image restoration model used in our comparisons, must be individually trained for each type of degradation.
>
> In practical applications, it is true that multiple degradations (such as rain and overexposure) can co-occur. While this is a significant challenge for single-model fusion, our proposed algorithm offers a lightweight and efficient solution, achieving a processing speed of 30 frames per second in real-time applications. Although some recent unified models, like DA-CLIP [1] and MPerceiver [2], aim to address multi-degradation issues, ***they require substantial computational resources and are not designed to handle complex combinations like de-raining and overexposure simultaneously.***
>
> **W5** `(2) About the Multi-Degradation Processing Capability of Proposed Method`
>
> In summary, while we recognize the importance of handling multiple degradations, it remains a challenging area that requires further exploration. ***Our current work focuses on providing a real-time, multi-modality fusion solution across complex scenarios with minimal parameters, which we believe represents a valuable contribution to the field.*** Addressing multiple degradations continues to be a challenge for the proposed real-time and lightweight model.
>
> ---
> [1] Luo Z,  et al. Controlling Vision-Language Models for Multi-Task Image Restoration. ICLR 2024.
>
> [2] Ai Y, et al. Multimodal Prompt Perceiver: Empower Adaptiveness Generalizability and Fidelity for All-in-One Image Restoration. CVPR 2024.

---

> > ### Comment · Reviewer_Eiih · 2024-11-29
> >
> > I appreciate the effort the authors have put in during the rebuttal period, but I still have some additional concerns regarding this paper:
> >
> > 1. In the authors' response, it is noted that the performance of EN is suboptimal. Does this imply that the proposed method is more focused on addressing image degradation rather than the fusion process itself, essentially repairing the degradation but neglecting the integration of source image information?
> >
> > 1. How can it be demonstrated that the poor performance of the "restoration + fusion" comparison method is due to the shortcomings of the fusion approach rather than the restoration module applied? If a sufficiently high-performance restoration module is used to fully recover the degraded image to its original state, would the fusion method still yield subpar results? The manuscript also mentions that the "restoration + fusion" approach could lead to error accumulation, further suggesting that the comparative experiments in the paper are not entirely fair.
> >
> > 1. I would like to know for the results in Table 2, when calculating metrics for overexposure and lighting scenarios, whether the reference images used were the original images or the original images with adjusted lighting? If it is the latter, this would indicate that an unfair comparison was made, as the images were artificially created by the authors, and the compared methods were not trained on such adjusted samples. Even if the comparison methods were applied directly to the adjusted original images, the fusion results may still be suboptimal, which further suggests that the benchmark used in this paper is unfair.
> >
> > 1. In addition to the custom dataset used in this study, please also compare the fusion results on standard datasets such as MSRS [1] and Roadsence[2]. Ideally, if the proposed method performs well on degradations like noise or rain, it should also perform excellently on the original images without such degradations. This is analogous to a mature image denoising algorithm, which should output the original clean image when the input is already clean.
> > [1] L. Tang, J. Yuan, H. Zhang, X. Jiang, and J. Ma, “Piafusion: A progressive infrared and visible image fusion network based on illumination aware,” Infromation Fusion.
> > [2] H. Xu, J. Ma, Z. Le, J. Jiang, and X. Guo, “Fusiondn: A unified densely connected network for image fusion,” AAAI 2020.

---

> > > ### Author Response · Authors · 2024-11-30
> > > **Response to Reviewer Eiih (C1-C3)**
> > >
> > > *We sincerely appreciate the valuable and constructive comments provided by the reviewer Eiih, which have significantly contributed to the improvement of our manuscript. Below, we present our detailed, point-by-point responses. Thank you for your time and effort in reviewing our work.*
> > >
> > > **C1**  `(1) About the Performance on EN Metric`
> > >
> > > Thank you for your question. First, while the EN metric for our method is not the highest, it is important to note that our EN scores are competitive and rank second across three scenes. As discussed in both the manuscript and our previous response, the top-ranked CoCoNet shows inflated EN scores due to pixel redundancy, which does not necessarily reflect better fusion quality. Additionally, when considering the 11 metrics we compared, our method consistently achieves the best overall performance across all scenes.
> > >
> > > **C1**  `(2) About the Focus on the Proposed Model`
> > >
> > > Second, the EN metric alone cannot determine whether the proposed algorithm focuses more on degradation removal or fusion. To provide a complete evaluation, we included comparisons in ideal scenes without degradation (***as shown in Tables 7 and 8 in the Appendix***). These results, based on the M3FD and LLVIP datasets, demonstrate that our method remains state-of-the-art in terms of fusion performance under non-degraded conditions.
> > >
> > > Lastly, while calculating metrics, we evaluate the fusion results separately with clean visible and infrared images. This process inherently assesses both the removal of degradation and the effectiveness of feature integration, showcasing that our method unifies these two tasks rather than focusing exclusively on one.
> > >
> > > **C2**  `(1) About the “Restoration + Fusion” Comparative Method`
> > >
> > > First, regarding the “restoration + fusion” comparison method, we believe it provides a reasonable and fair benchmark. This approach is currently the only way to enable fusion algorithms that are not inherently designed for complex scenarios to handle such cases. It has also been adopted in other studies, such as Text-IF [1], TFS-Dif [2], MURF [3] and Mdbfusion [4], further supporting its validity as a comparison strategy. While we acknowledge that error accumulation can occur in the fusion stage, this limitation highlights the need for a unified framework like ours that avoids the sequential processing pitfalls of separate restoration and fusion stages.
> > >
> > > *[1] Xunpeng Yi, et al. Text-IF: Leveraging Semantic Text Guidance for Degradation-Aware and Interactive Image Fusion. CVPR 2024.*
> > >
> > > *[2] Yusen Xu, et al. Simultaneous Tri-Modal Medical Image Fusion and Super-Resolution using Conditional Diffusion Model. MICCAI 2024.*
> > >
> > > *[3] Han Xu, et al. Murf: Mutually reinforcing multi-modal image registration and fusion. TPAMI, 2023.*
> > >
> > > *[4] Jun Chen, et al. Mdbfusion: A visible and infrared image fusion framework capable for motion deblurring. ICIP 2024.*
> > >
> > > **C2**  `(2) About the Use of the High-Performance Restoration Model`
> > >
> > > We acknowledge that utilizing a highly capable restoration module capable of fully recovering degraded images to their original state could improve the performance of subsequent fusion algorithms. However, this approach overlooks the trade-off between performance and efficiency, as our objective is to develop an algorithm suitable for real-time applications. To strengthen the experimental evaluation, we employ a new recovery model, DA-CLIP [5], as a pre-processing algorithm for degraded images. The quantitative comparison results are presented in the following table.
> > >
> > > *[5] Ziwei Luo, et al. Controlling Vision-Language Models for Multi-Task Image Restoration. ICLR 2024.*
> > >
> > > **Tab 1.**  *Quantitative Comparison on Rain Scenes.*
> > >
> > > | Methods   | Restoration  | $Q_{MI}$   | $Q_{NCIE}$  | $Q_{M}$ | $Q_{P}$         | $Q_{S}$    | $Q_{CB}$   | $SSIM$   | $PSNR$    | $SCD$    |
> > > |-----------|-------------|--------|--------|--------|--------|--------|--------|--------|---------|--------|
> > > | CDDFuse   | DA-CLIP     | **0.3903** | 0.8035 | 0.4372 | 0.1441 | **0.7941** | 0.4241 | 0.2813 | 18.8025 | 1.3549 |
> > > | IGNet     |      DA-CLIP       | 0.2619 | 0.8033 | 0.3996 | 0.1619 | 0.7105 | 0.4413 | **0.3084** | 18.9636 | 1.3743 |
> > > | Proposed  | w/o         | 0.2990 | **0.8040** | **0.4821** | **0.1622** | 0.7857 | **0.4494** | 0.2963 | **19.0249** | **1.3798** |
> > >
> > > **C3**  `Clarification on Quantitative Comparisons`
> > >
> > > In Table 2 of the manuscript, the overexposed images used in our experiments are synthetically generated, each paired with a corresponding GT. This approach is necessary because there is currently no large-scale dataset of paired multimodal images specifically for overexposure scenarios. The GT images serve as the reference for calculating the metrics, rather than adjusted original images.
> > >
> > > We would like to emphasize that we took great care to ensure the fairness of our experiments. Both quantitative comparisons and qualitative evaluations were designed to provide a level playing field for all methods.

---

> ### Author Response · Authors · 2024-11-30
> **Response to Reviewer Eiih (C4)**
>
> **C4**  `(1) About Experiments on Standard Datasets`
>
> Thank you for your question. In addition to the experiments conducted on synthetic degraded datasets, ***our study includes comparative experiments on standard datasets without degradation***, such as M3FD and LLVIP. These results are presented in Tables 7 and 8 in the Appendix. To further address your suggestion, we have now included quantitative comparison results on the MSRS and RoadScene datasets, which are provided in the following table.
>
> **Tab 2.**  *Quantitative Comparison on MSRS Dataset.*
>
> | Methods    | $Q_{MI}$   | $Q_{NCIE}$  | $Q_{M}$ | $Q_{P}$         | $Q_{S}$    | $Q_{CB}$   | $SSIM$   | $PSNR$    | $SCD$    |
> |------------|-------|--------|------------|------------|-------|-------|--------|---------|--------|
> | CoCoNet    | 0.3361| 0.8063 | 0.2074     | 0.3505     | 0.5759| 0.4180| 0.3825 | 9.2917  | 1.2966 |
> | Text-IF    | 0.8161| 0.8342 | **1.5393** | **0.6774** | 0.8752| 0.5471| 0.5955 | 11.1486 | 1.4670 |
> | CDDFuse    | **0.8484**| 0.8373 | 0.6914     | 0.6721     | 0.8140| 0.4643| 0.5471 | 10.9534 | 1.1957 |
> | DeFusion   | 0.4926| 0.8084 | 0.4916     | 0.3980     | 0.8496| 0.5186| 0.4611 | 17.9168 | 1.2925 |
> | IGNet      | 0.3029| 0.8047 | 0.4135     | 0.3324     | 0.4276| 0.3978| 0.3795 | 12.8023 | 1.4806 |
> | LRRNet     | 0.5168| 0.8121 | 0.5770     | 0.5446     | 0.8165| 0.4717| 0.5636 | 11.5010 | 0.7263 |
> | TGFuse     | 0.6349| 0.8202 | 0.6883     | 0.6289     | 0.8319| 0.4819| 0.5537 | 10.8003 | 1.2789 |
> | Proposed   | 0.7492| **0.8417** | 0.7844 (2) | 0.6737 (2) | **0.8793**| **0.5565**| **0.6987** | **18.0030** | **1.6095** |
> ---
>
> **Tab 3.**  *Quantitative Comparison on RoadScene Dataset.*
>
> | Methods    | $Q_{MI}$   | $Q_{NCIE}$  | $Q_{M}$ | $Q_{P}$         | $Q_{S}$    | $Q_{CB}$   | $SSIM$   | $PSNR$    | $SCD$    |
> |------------|-------|--------|------------|------------|-------|-------|--------|---------|--------|
> | CoCoNet    | 0.3561| 0.8061 | 0.3223     | 0.3031 | 0.6510 | 0.4892| 0.3863 | 12.4883      | 1.7926 |
> | Text-IF    | 0.3475| 0.8058 | 0.3553     | 0.2019 | 0.6257 | 0.4650| 0.2867 | 13.6850      | 1.3116 |
> | CDDFuse    | 0.4208| 0.8073 | 0.4231     | 0.3704 | 0.7547 | 0.4873| 0.4780 | 14.0086      | 1.7118 |
> | DeFusion   | 0.4304| 0.8076 | 0.3860     | 0.3529 | 0.7753 | 0.5084| 0.4588 | **16.1255**  | 1.3210 |
> | IGNet      | 0.2417| 0.8042 | 0.2508     | 0.0615 | 0.3954 | 0.4364| 0.1090 | 10.1571      | 1.0344 |
> | LRRNet     | 0.3929| 0.8065 | 0.3610     | 0.2444 | 0.5936 | **0.5096**| 0.3295 | 11.8040  | 1.5690 |
> | TGFuse     | 0.3725| 0.8062 | **0.6462** | 0.4086 | 0.7764 | 0.4555| 0.4604 | 14.4941      | 1.4181 |
> | Proposed   | **0.6285**| **0.8170** | 0.6343 (2) | **0.5556** | **0.7796** | 0.4598| **0.4821** | 15.0375 (2) | **1.8601** |
>
> **C4**  `(2) About Model Performance in Different Scenes`
>
> Additionally, to evaluate the performance of our model on clean images, we compared fusion results from degraded images with those from clean images using the model trained on noise and rain scenes respectively (***As shown in Table 4 and Table 5***). The results demonstrate that the proposed algorithm maintains high fusion quality even in non-degraded conditions, underscoring its generalizability and robustness.
>
> **Tab 4.**  *The proposed method employs quantitative comparison experiments using different source images with weight trained on noise scenes.*
>
> | Sources    | $Q_{MI}$   | $Q_{NCIE}$  | $Q_{M}$ | $Q_{P}$         | $Q_{S}$    | $Q_{CB}$   | $SSIM$   | $PSNR$    | $SCD$    |
> |---------|-------|-------|-------|-------|-------|-------|-------|---------|--------|
> | Noise (Train in Noise)   | 0.4504 | 0.8080 | 0.4072 | 0.8151 | 0.2158 | 0.4609 | 0.3709 | 15.3349 | 1.4868 |
> | Clean (Train in Noise)  | **0.4816** | **0.8096** | **0.5025** | **0.8364** | **0.2702** | **0.4702** | **0.3750** | **16.9659** | **1.6525** |
> ---
>
> **Tab 5.**  *The proposed method employs quantitative comparison experiments using different source images with weight trained on rain scenes.*
>
> | Sources    | $Q_{MI}$   | $Q_{NCIE}$  | $Q_{M}$ | $Q_{P}$         | $Q_{S}$    | $Q_{CB}$   | $SSIM$   | $PSNR$    | $SCD$    |
> |---------|-------|-------|-------|-------|-------|-------|-------|---------|--------|
> | Rain  (Train in Rain)  | 0.2990 | 0.8040 | 0.4821 | 0.1622 | 0.7857 | 0.4494 | 0.2963 | 19.0249 | 1.3798 |
> | Clean (Train in Rain)   | **0.4205** | **0.8054** | **0.5491** | **0.2707** | **0.8416** | **0.4682** | **0.4195** | **19.8320** | **1.4366** |
>
> ---
> Finally, thank you for your valuable feedback, which has helped us further strengthen our experimental analysis. If you believe our responses have adequately addressed your concerns, we would sincerely appreciate your consideration in raising your rating. Thank you again for your time and thoughtful feedback.

---

### Author Response · Authors · 2024-12-04
**To PCs, SACs, ACs, and Reviewers**

Dear PCs, SACs, ACs, and Reviewers,

We would like to thank you for your valuable feedback and insightful reviews, which have greatly contributed to improving the paper. Below, we summarize the key strengths of our work as highlighted in the reviews, which we believe support the merit and potential impact of our contribution:

**(1)	Innovative Methodology**

Our work introduces a novel method that effectively addressing image fusion in complex scenes such as noise, low light, and adverse weather conditions. ***Reviewer 5BEs*** emphasized that this is the first study to address complex scene fusion in the frequency domain.

**(2)	Efficiency and Practicality**

The lightweight design enables real-time processing, requiring just 0.033 seconds for a 640×480 image. This was noted by multiple reviewers (***Reviewers Eiih, Pge1, and 5BEs***) as a significant advantage for real-world applications like autonomous driving and industrial monitoring.

**(3)	Experimental Rigor and Performance**

Extensive experiments validate the robustness and superiority of our method over state-of-the-art approaches, as highlighted by ***reviewers Pge1 and 5BEs***.

**(4)	Clarity and Structure**

***Reviewers xvYN and 21oJ*** appreciated the clear writing, logical organization, and detailed explanation of the methodology.

---

**The main additions are as follows:**

-	Added a detailed explanation of the rationale for using the MSEC algorithm. ***(See our response to Reviewer Eiih W3 (3))***

-	Elaborated on the comparison with "image restoration + image fusion" approaches. ***(See our response to Reviewer Eiih C2 (1))***

-	Added experiments with new evaluation metrics, including the no-reference metric $EN$ and the reference-based metric $Q_{P}$. ***(See our response to Reviewer Eiih W3 (1) and (2))***

-	Conducted additional analyses, including computational efficiency comparisons of "restoration + fusion" methods, tests with DA-CLIP as a pre-restoration algorithm, and experiments on standard datasets (MSRS and RoadScene). ***(See our response to Reviewer xvYN W2 and Eiih C2 (2) and C4 (1))***

-	Included tests of the proposed algorithm trained on degraded scenes and evaluated on normal scenes. ***(See our response to Reviewer Eiih C4 (2))***

-	Added comparisons with four state-of-the-art methods: MURF, EMMA, DDFM, and SegMiF. ***(See our response to Reviewer xvYN W3)***

We understand the time constraints and workload that reviewers and Area Chairs face, and we deeply appreciate the effort in evaluating our work. Your feedback is invaluable in improving our research.

Best regards,

Authors of Submission 6459

---

### Meta-Review · Area_Chair_TBRS · 2024-12-20

**Metareview:**

This paper addresses the limitations of current infrared and visible image fusion (IVIF) methods when applied to complex scenes with interferences such as rain, noise, and low-light conditions. The authors propose a unified IVIF framework that integrates image restoration and fusion without the need for pre-processing. Using Fourier domain techniques, the framework captures amplitude information from infrared and visible images and combines these with spatial domain data to maintain clarity and reduce interference.
This paper receives significant dispersed scores among the reviews. Reviewers with positive comments believe that the paper is well-described, with clear motivation and good experimental results. However, Reviewer Eiih with negative comments thinks that the methodology has flaws in experimental setup and model generalization, while Reviewer xvYN feels that the method of exchanging amplitude and phase lacks novelty. Thus, this paper is slightly below the acceptance threshold.

**Additional Comments On Reviewer Discussion:**

The author provided detailed answers and sufficient experimental results in rebuttal. Reviewers 5BEs and 21oJ both feel that their questions were well addressed and subsequently raised their scores. However, the author did not completely convince reviewer Eiih, who remains concerned about the effectiveness of the paper's methodology and the fairness of the experimental comparisons, maintaining the reject score.

---

### Decision · Program_Chairs · 2025-01-22

Reject